# Episomal and integrated hepatitis B transcriptome mapping uncovers heterogeneity with the potential for drug-resistance

James M. Harris [1], James Lok [2], Nadina Wand [1], Andrea Magri [1], Senko Tsukuda[3], Yanxia Wu [4], Esther Ng[5], Daisy Jennings[1], Badran Elshenawy[1], Peter Balfe [1] & Jane A. McKeating [1,3] ✉

Hepatitis B virus (HBV) is a small DNA virus that establishes chronic infection and drives progressive liver disease and cancer; presenting a global health problem with more than 250 million infections. HBV replicates via an episomal covalently-closed-circular DNA (cccDNA) and integrated viral DNA fragments are linked to carcinogenesis. Current treatments only suppress HBV replication and there is a global initiative to develop genome targeting therapies, including siRNAs, antisense oligonucleotides and epigenetic modifiers specific for HBV cccDNA. However, our knowledge of the cccDNA and integrant transcriptomes is confounded by overlapping viral RNAs. Using targeted long-read sequencing we mapped the HBV transcriptome in liver biopsies from eleven treatment naïve patients. Probe enrichment yielded robust sequencing libraries and identified cccDNA-derived genomic and sub-genomic transcripts, and a repertoire of previously uncharacterised spliced, truncated and chimeric viral RNAs. Assigning viral transcripts to their respective DNA templates revealed differential promoter activity in cccDNA and integrants, with implications for the efficacy of epigenetic modifiers. Integrant-derived transcripts showed vast diversity in the viral-host junctions, posing a challenge for current nucleotide-targeting therapies. cccDNA was a source of genetic polymorphism, with distinct viral lineages present in the surface antigen encoding region, providing an insight into hepadnavirus evolution during chronic infection.

Hepatitis B virus (HBV) is a small DNA virus and the prototypic member of the *hepadnaviridae* family, presenting a global health problem with >250 million infections worldwide. Chronic hepatitis B (CHB) is associated with a progressive liver disease including fibrosis, cirrhosis and hepatocellular carcinoma (HCC), accounting for 1 million deaths annually[1]. HBV exists as ten different genotypes (A to J)

based on overall genetic distances of 4–7.5% with multiple subtypes. Many genotypes are geographically localised such that racial, environmental and other confounding factors complicate the association of genotype with disease progression[2-4]. In the absence of curative therapies viral infection can be life-long. Peripheral biomarkers of viral activity and liver damage are used to categorise CHB disease

[1]Nuffield Department of Medicine, University of Oxford, Oxford, UK. [2]Institute of Liver Studies, Kings College London, London, UK. [3]Chinese Academy of Medical Sciences Oxford Institute, University of Oxford, Oxford, UK. [4]OMICS Technology Platforms, Centre for Human Genomics, Nuffield Department of Medicine, University of Oxford, Oxford, UK. [5]Kennedy Institute of Rheumatology, University of Oxford, Oxford, UK. ✉e-mail: jane.mckeating@ndm.ox.ac.uk

stages but are likely to oversimplify the complex virus-host interplay in the liver[5].

HBV exclusively infects hepatocytes, with tropism defined by the expression of the viral entry receptor sodium taurocholate co-transporting polypeptide (NTCP)[6,7]. The template for viral replication is an episomal 3.2 kb covalently closed circular DNA (cccDNA) that persists in the nucleus of infected cells as a mini chromosome complexed with histones. Consequently, viral gene expression is dependent on host RNA polymerase II and transcription factors[8]. Viral transcription is orchestrated by four promoters (Core, Sp1, Sp2 and X) along with two enhancer elements (EnhI/II), that regulate six major RNAs, each with a unique 5' transcription start site (TSS) and a shared 3'polyadenylation signal (PAS)[9]. Our understanding of HBV promoter activity in clinical samples is limited. HBV replicates via polymerase (Pol) mediated reverse transcription of the pregenomic RNA (pgRNA) to generate a relaxed circular DNA (rcDNA)[10,11]. Aberrant replication can occur during reverse transcription, generating a double stranded linear (dsl) DNA that can be imported into the nucleus where it can form an incomplete cccDNA[12] or integrate into the host chromosome via non-homologous end joining[13,14]. Integration is a replicative dead end for the virus, as linearisation separates the Core and polymerase open reading frames (ORFs) from the Core promoter, precluding transcription of genomic length transcripts. Integrated HBV DNA (iDNA) was first identified in the context of HCC, which led to the belief that integration drives oncogenesis. Integration introduces viral promoters into the host genome that can be activated by hepatic transcription factors, with the potential to modulate host gene expression, providing a mechanistic link between iDNA and tumorigenesis[15]. Whilst structural maintenance of chromosome 5/6 (Smc5/6) has been identified as a fundamental repressor of cccDNA[16], to date our understanding of other host pathways that suppress HBV iDNA is limited.

The compact viral genome gives rise to overlapping RNAs, posing a significant challenge to mapping the HBV transcriptome. Whilst RT-qPCR, digital-droplet PCR, next generation sequencing and 5'RACE technologies have all been applied to HBV, the low abundance of viral reads in clinical samples compromises their utility[17–22]. Furthermore, the fragmentation and selective amplification required for short-read sequencing precludes the resolution of full length RNAs. To address this gap in knowledge, we employed a targeted long-read sequencing approach to enrich HBV RNAs using a panel of biotinylated oligonucleotide probes spanning the viral genome, resulting in an approximate 700-fold increase in their abundance[23,24]. Although HBV replication is defined by the full length RNAs, our earlier in vitro study showed these transcripts are rare and represented ~8% of the viral transcriptome[24], their abundance in clinical samples is not well studied. In contrast the Sp1 and Sp2 promoters direct synthesis of transcripts encoding the viral surface glycoproteins (HBs) and constitute up to ~87% of the viral transcriptome in vitro[24]. iDNA has been implicated in the constitutive expression of HBs and we and others have noted that they are the dominant source of viral RNAs in the HBe antigen (HBeAg) negative stage of disease[25–27]. HBs is a fundamental component of the infectious viral particle; with reported roles as an immune decoy resulting in T cell exhaustion, and in the onset and development of HCC[28–30]. A small number of chronically infected subjects spontaneously resolve infection and this associates with undetectable peripheral HBs, consequently this biomarker is a target endpoint for many clinical trials as a correlate of functional cure[31,32]. Available therapies do not target cccDNA and there is a global effort to develop curative therapies with novel siRNAs, antisense oligonucleotides (ASOs) and epigenetic modifiers currently in development[33]. However, the dual source of HBV transcripts provides a hurdle to functional cure, with ongoing iDNA transcription acting as a reservoir for constitutive antigen expression[33,34]. A greater understanding of cccDNA and iDNA transcription and their contribution to viral biomarkers is essential to validate these new antiviral and immuno-modulatory therapies.

In this study, we explored the hepatic HBV transcriptome using targeted long-read sequencing in a cohort of treatment naïve patients. Using RNA isolated from liver biopsies, we sequenced genomic and subgenomic RNAs and profiled the activity of the viral promoters in both viral templates. We characterised the heterogeneity of integrant derived transcripts, noting diversity in the viral-host junctions generated during integration, with truncated iDNA presenting a challenge for current nucleotide-targeting therapies. Importantly, we identified cccDNA as a source of genetic polymorphism, observing distinct viral lineages present in the HBs encoding region, providing an insight into hepadnavirus evolution during chronic infection.

## Results

### Defining the HBV transcriptome

Liver biopsy samples from 46 CHB subjects in HBeAg positive or negative phases of disease with variable liver fibrosis and inflammatory scores, summarised in Supplementary Fig. 1A, were assessed for total HBV transcripts and pgRNA levels[17]. Among the HBeAg positive samples we observed a 2–3 log range in the number of total HBV transcripts, with a similar variation in pgRNA (Fig. 1A). In contrast, we noted a 6–7 log range in the abundance of total HBV transcripts in HBeAg negative subjects, with significantly lower levels of pgRNA, ranging from $10^7$ copies to undetectable. Estimating the amount of pgRNA relative to total HBV RNA in HBeAg positive patients showed that genomic RNA was ~38% of the total, whereas this was significantly reduced to ~4% in HBeAg negative disease. As the majority of chronically infected patients worldwide are HBeAg negative we selected eleven subjects in this disease phase infected with the same genotype D virus and who were treatment naïve at the time of biopsy (Supplementary Fig. 1B). Viral RNAs were enriched using a previously reported probe-based capture technique[24] and sequenced using a long-read method (PacBio). To allow comparison between patient samples the viral sequences were aligned, using minimap2, to a referent HBV genotype D 3.5 kb overlength genome (NC_003977.2). The circular HBV genome has a unique EcoRI restriction site that is frequently annotated as the first base for historical reasons (Supplementary Fig. 2A). However, when mapping the viral transcriptome it is logical to designate the first nucleotide of the preCore (pC) transcript as the first base, with the EcoRI site at position 1658 and the 3' terminal Poly Adenylation Site (PAS) at position 3581[35] (Fig. 1B). In addition, we sequenced RNA from HepG2-NTCP hepatoma cells infected with the HBV genotype D molecular clone (NC_003977.2) (Supplementary Fig. 2C). We sampled these in vitro infections after 6 days to align with previous studies showing that HBV transcription is largely driven from cccDNA and integration events are rare at this time[24,36]. Up to half of the sequences obtained were of viral origin with a median read count of 27,956 in the clinical samples and 77,841 in the experimental infections; ~90% of sequences aligned to the 3' end of the linear genome (Fig. 1C, Supplementary Fig. 2B–C, Supplementary Data 1). We noted a significant association between the HBV reads, expressed as transcripts per million (TPM), and RT-qPCR measurement of the total viral RNAs, suggesting unbiased probe enrichment across the samples (Supplementary Fig. 2D).

HBV cccDNA has four promoters and two enhancer regions that direct the synthesis of six major transcripts. The Core promoter regulates the transcription of pC and pg full-length RNAs; the Sp1 and Sp2 promoters drive the preS1, preS2 and S RNAs that encode HBs and the X promoter directs HBx expression (Fig. 1B). To profile Core promoter activity, we quantified pC long and short isoforms (pC-L, pC-S) and pgRNA initiating at reported TSS (Fig. 1D)[19]. A significant number of reads with previously unreported TSS were identified and classified as group 1 Core-associated transcripts (C-AT1: TSS range 13-211) or group 2 (C-AT2: TSS range 280–1196) (Fig. 1D, Supplementary Fig. 2E). Core promoter mutations and deletions are widely reported in the

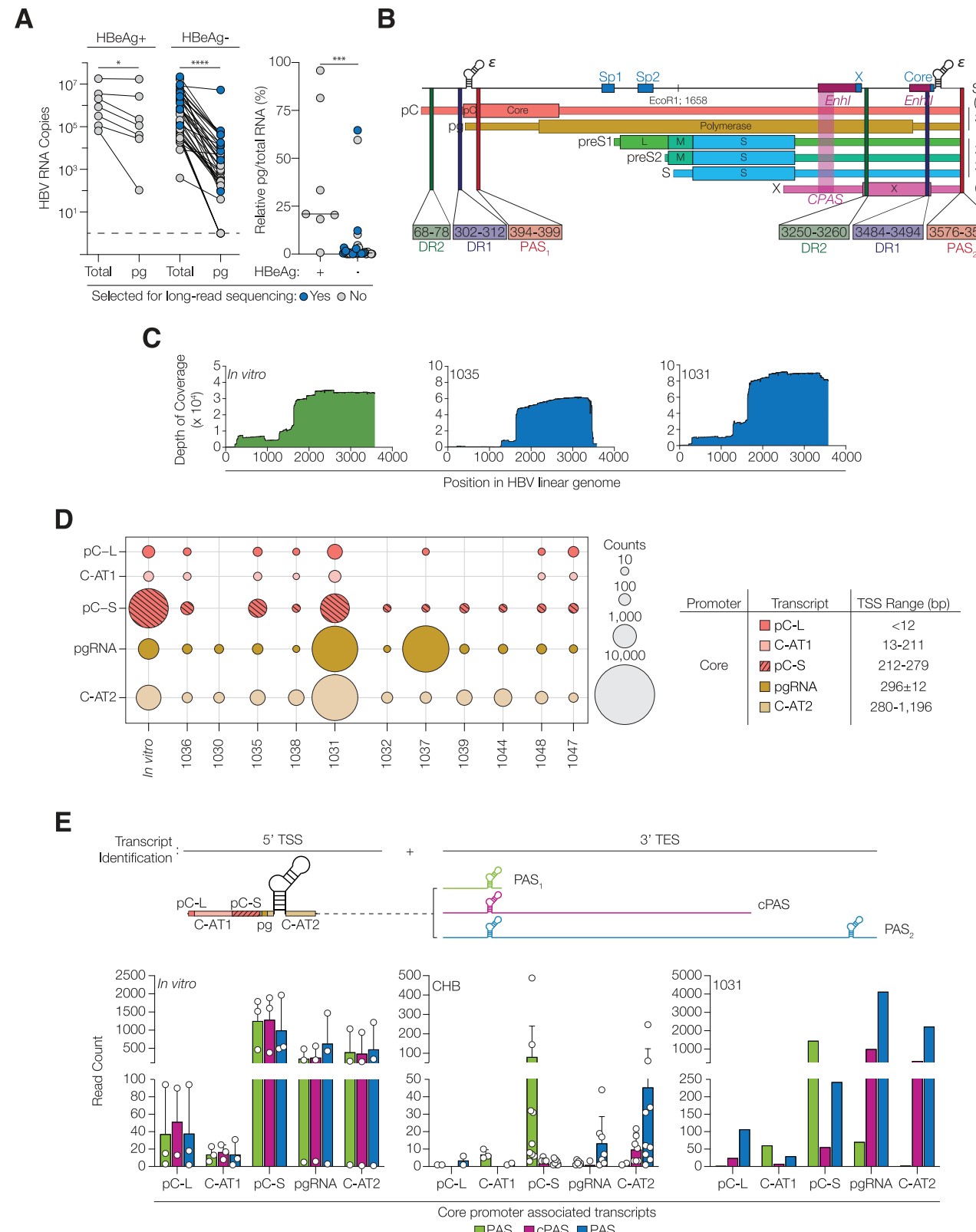

literature[37], though their functional impact is not fully understood. Nine of the eleven patients had a consensus genotype D promoter sequence, with two samples (1035 and 1037) having deletions in this region[27]. However, Core-directed transcripts were found in all samples, suggesting active promoters in all biopsies.

The genesis of full-length (3.5 kb) RNA from a circular 3.2 kb DNA template requires the host transcription machinery to read through the first PAS (PAS$_1$ bp 394-399) and terminate at the second PAS (PAS$_2$ bp 3575–3581) (Fig. 1B). Long-read sequencing identifies both the 5' (TSS) and 3' (transcript end site, TES) termini of the RNAs, enabling the detection of abortive transcription. Assessing the 3' termini of Core promoter directed RNAs identified 3 conserved TES: transcripts that terminated at PAS$_1$ or PAS$_2$ (TATAAA), together with a significant number terminating at ~ 2600 bp that we defined as a cryptic PAS

**Fig. 1 | Ascribing HBV transcripts to the Core promoter. A** RNAs were extracted from liver biopsies from 46 patients and HBV transcript abundance estimated by RT-qPCR for all HBV RNAs (Total HBV RNA) or the unique pgRNA locus. Copy numbers were enumerated from a standard curve and adjusted relative to RPLP0 expression (Delta Ct method) and the ratio of pgRNA to total RNA expressed as a percentage, with the threshold of detection shown. Differences were assessed with Wilcoxon matched-paired signed rank test ($p = 0.0156$ *, $p < 0.0001$ ****; two-tailed) and Mann–Whitney $U$ tests ($p = 0.0002$ ***; two-tailed). **B** Schematic showing the major HBV transcripts, with unique 5′ TSS and conserved 3′ TES at $PAS_2$. The protein coding regions, 4 promoters, 2 Enhancers, Direct repeats 1/2 (DR1/2), and the EcoRI restriction site are depicted. **C** RNA from de novo infected HepG2-NTCP cells and from 11 HBeAg negative liver biopsies were enriched for HBV transcripts using a panel of oligonucleotide capture probes[24]. Reads were mapped to an overlength HBV genome (D3L), with the depth of coverage at each position shown for a representative in vitro infection (1014) and 2 biopsy samples (1035 and 1031). **D** The TSS of HBV RNAs were annotated as previously defined[24]. Reads initiating upstream of Sp1 were defined as pC-L, pC-S or pgRNA, with transcripts initiating outside these boundaries identified as Core-associated transcript group 1 or 2 (C-AT1 or C-AT2). Read counts are depicted as bubbles, where size is proportional to number of reads. **E** A schematic showing the 5′ and 3′ termini used for transcript assignment. The 3′ sequence identified transcripts terminating at $PAS_2$ together with abortive transcripts terminating at $PAS_1$ during the first circuit of cccDNA (position 400) or at a cryptic PAS (position 2600). Read counts are shown for the in vitro infection samples ($n = 3$; mean + range) and CHB patient samples ($n = 11$; mean + SD) with 1031 classified as an outlier. See Supplementary Fig. 1, 2. Source data are provided as a Source Data file.

(cPAS) (Fig. 1E). Analysing sequences around this termination site did not identify any hexameric PAS motifs previously reported in the human transcriptome[38]. In the experimentally infected samples, we noted that ~40% of pgRNA terminated at this cPAS and do not encode an intact Pol ORF. Although these transcripts are unlikely to be packaged or reverse transcribed, they encode an intact core ORF, suggesting a potential mechanism to regulate core and polymerase expression (Fig. 1E). This contrasts with the clinical samples, where the majority of Core-associated transcripts terminated at $PAS_2$, suggesting differential regulation of the Core promoter in vivo. Sample 1031 was atypical and showed a complex pattern of TES, with many under length pC-S, pgRNA and C-AT2 transcripts. This high frequency of shorter transcripts poses an issue for PCR-based approaches that quantify pgRNA[39,40] and highlights the diverse repertoire of viral RNAs.

Having profiled the Core promoter, we sought to evaluate the contribution of the Sp1, Sp2 and X promoters to the viral transcriptome. In all clinical samples, Sp2 driven preS2 transcripts were the most abundant, with preS1, S and X RNAs readily detectable, however 37% of viral sequences did not initiate at previously reported TSS (Fig. 2A, Supplementary Data 1). The 5′ termini of these 'orphan' RNAs showed a complex pattern, with several conserved initiation sites among the clinical samples, consistent with adventitious transcript initiation (Fig. 2B). Elucidating the TSS of the viral RNAs allowed us to identify their cognate promoters (Supplementary Figs. 2E, 3A). The Sp1 promoter associated transcripts (Sp1-AT) initiated over a 375 nucleotide range (1197–1572) with evidence of two conserved TSS at positions 1245–1255 and 1300–1309. We also observed new TSS in the Sp2-AT region, with four conserved initiation sites at 1581–1591; 1610–1622; 1695–1702 and 1704–1715, upstream of the S-HBs start codon (Fig. 2B). Over 90% of transcripts initiating down-stream of the X promoter (X-AT) encoded a complete HBx ORF and constituted up to 12% of the viral transcriptome. In contrast, analysing the TSS of the three most abundant host transcripts, Albumin (*ALB*), Fibrinogen gamma chain (*FGG*) and Haptoglobin (*HP*) showed that initiation occurred within a narrow, 24 bp window. *ALB* had the most conserved TSS with 93% of transcripts starting in this window, whereas *FGG* (84%) and *HP* (85%) TSSs were less conserved, reflecting alternative isoform expression (Fig. 2C).

Given the significant number of abortive Core-associated transcripts, we were interested to study the 3′ termini of Sp1, Sp2 and X promoter associated RNAs. In the experimental infections the majority of transcripts terminated at $PAS_2$ with a minority terminating at the newly identified cPAS. cPAS usage was frequently observed in the Core promoter derived transcripts, suggesting that cccDNA topology or chromatinization may define this process (Fig. 2D). In clinical samples most transcripts terminated ~200 base pairs upstream of $PAS_2$ (Fig. 2D, Supplementary Fig. 3B), consistent with transcription from integrated viral genomes. Our protocol allowed us to sequence the polyA tails of transcripts and reassuringly >99.99% were polyadenylated with an average length of 30–50 bases (Supplementary Fig. 3C). Of note, Core-promoter associated RNAs had longer polyA tails, with C-AT1 and pC-S RNAs showing an average 139 and 164 bases, respectively

(Supplementary Fig. 3C)[24]. These data highlight the complex pattern of RNA initiation from the HBV promoters and the high frequency of truncated transcripts in the biopsy samples.

## Ascribing HBV transcripts to cccDNA or iDNA

HBeAg negative disease is associated with reduced cccDNA levels and lower inferred transcriptional activity[26,27], however, there is limited quantitative information from liver biopsies. We previously published low to undetectable levels of cccDNA in this cohort[27] with only five of the eleven patients sequenced in this study having detectable cccDNA using a digital droplet PCR method (Supplementary Fig. 1B). Accurate measurement of cccDNA activity would facilitate the stratification of patients for new treatments targeting the HBV epigenome. Most RNAs transcribed from cccDNA (ccc-RNA) will terminate at $PAS_2$ which is lost during dslDNA genesis[25,27,41,42], consequently RNAs lacking this motif are defined as integrant derived (id-RNA). The majority (median 98%) of viral RNAs obtained from the biopsies were derived from integrants as they lacked the $PAS_2$ sequence and terminated upstream of the 3′DR1 motif. This contrasts with the short-term experimental infections where only ~2% of transcripts were classified as id-RNAs (Fig. 3A). Both clinical and experimental samples contained transcripts terminating at the newly identified cPAS that could originate from either cccDNA or iDNA, these were classified as 'unassigned' (Fig. 3A). One biopsy sample, 1031, had a high frequency (61%) of ccc-RNA, reflecting an active cccDNA reservoir.

Assigning viral RNAs to their DNA templates allowed us to compare promoter activity in episomal and integrated genomes (Fig. 3B). Consistent with the switch to iDNA as the major source of RNAs in this disease phase[26,27], we found that several samples (1030, 1032, 1037, and 1044) had fewer than 100 ccc-RNAs and were removed from subsequent analyses (Fig. 3). Among the clinical samples the id-RNA promoter profiles were similar, with Sp2 driven transcripts comprising >70% of the sequences. In contrast, the ccc-RNA populations were more variable with notable differences in the relative contribution of the Sp1 and Sp2 promoters and low Core promoter activity in many samples. HBx promoter was active in both cccDNA and iDNA and directed up to 12% of the viral transcriptome (Fig. 3B). We observed unique patterns of spliced viral RNAs from each template, with iDNA giving rise to a complex repertoire, noting SP14 and pSP12 transcripts as the most abundant iDNA associated spliced RNAs (Supplementary Fig. 4A, Supplementary Data 1). Higher levels of the SP1 spliced RNA were found in samples with detectable pgRNA (Supplementary Fig. 4B), consistent with previous studies[42]. These data highlight the variable promoter activity in cccDNA compared to iDNA, consistent with different epigenetic pathways and host restriction factors regulating these two templates.

## In-depth characterisation of integrant derived RNAs

To date, the HBV integrant associated transcriptome in vivo is understudied. HBV integration disrupts the Core promoter and precludes the transcription of genomic length RNAs but maintains the Sp1, Sp2 and X promoters. As integrants lack the HBV encoded $PAS_2$,

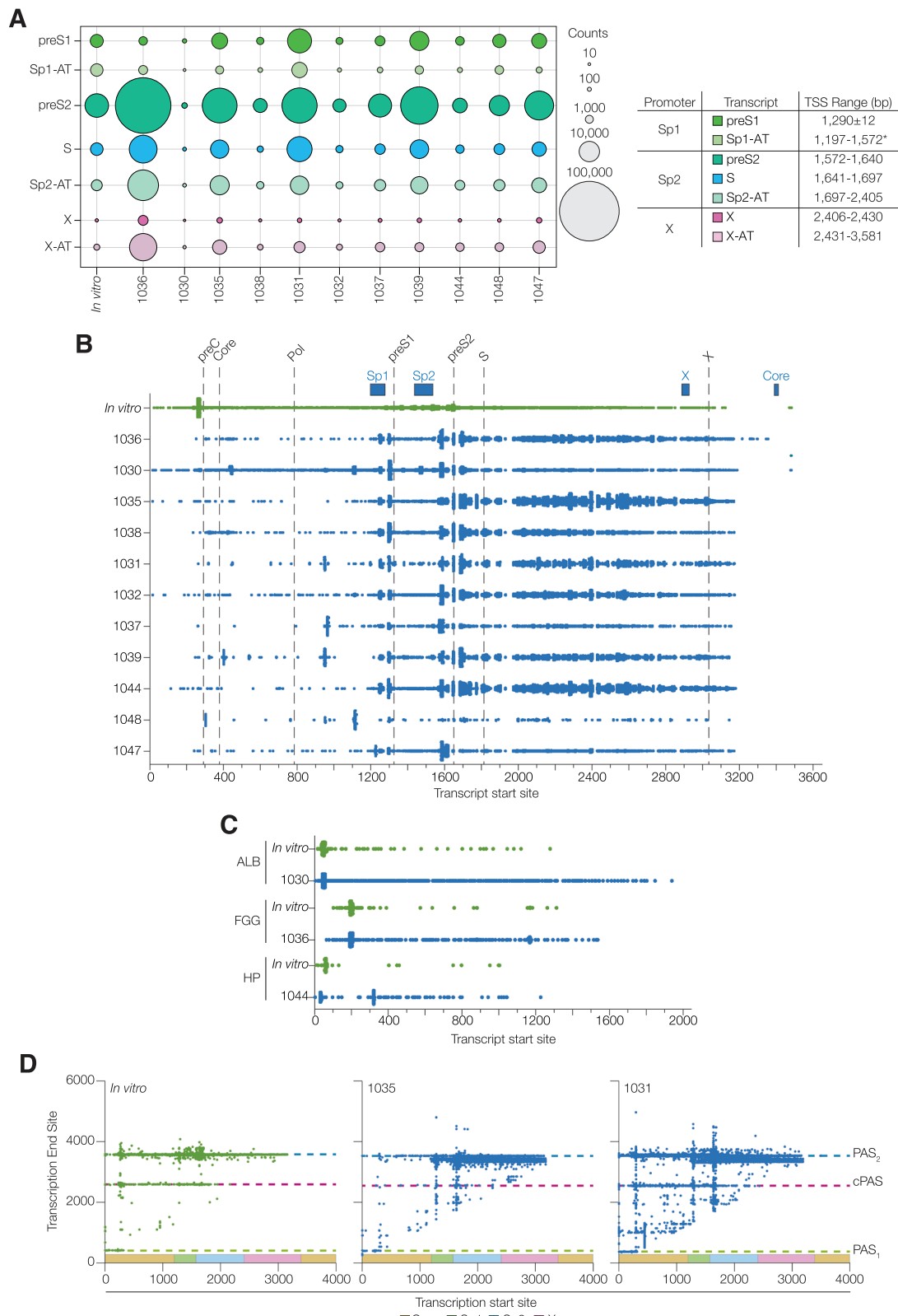

**Fig. 2 | Identification of HBV Transcript Start (TSS) and End (TES) coordinates.**
**A** TSS coordinates (bp) of the canonical (preS1, preS2, S and X) and promoter associated transcripts (Sp1-AT; Sp-AT and X-AT) are shown for the in vitro and clinical samples, with read counts depicted as a bubble plot. **B** TSS of the orphan transcripts are depicted on a linearised genome, annotated with the viral promoters (blue boxes) and the ATG start codons (dashed lines). In vitro and clinical samples are shown, where each symbol represents an individual transcript. **C** TSS of

selected host transcripts (Albumin (*ALB*), Fibrinogen Gamma Chain (*FGG*) and Haptoplatin (*HP*)) from representative in vitro and clinical samples (1014, 1030, 1036 and 1044), where symbols represent a single transcript. **D** Scatter plots showing the transcription start and end sites for all RNAs from representative in vitro and clinical samples (1014, 1035, 1031). PAS$_1$, cPAS, and PAS$_2$ positions are depicted as dashed lines, with promoter regions shaded. See Supplementary Fig. 3 for all samples. Source data are provided as a Source Data file.

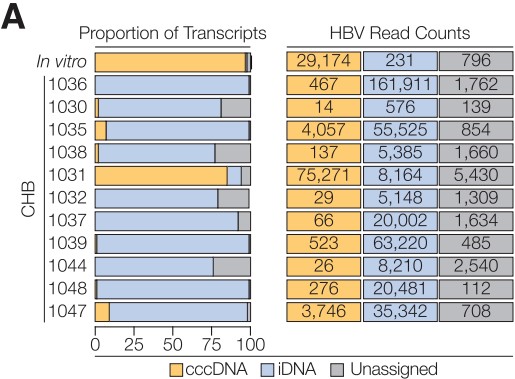

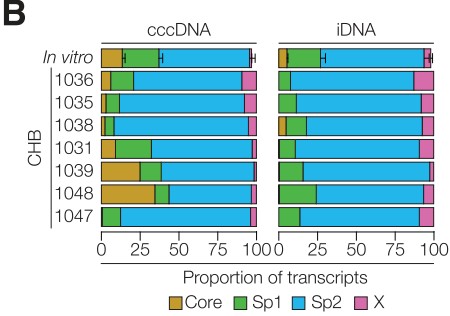

**Fig. 3 | Assigning cccDNA or iDNA transcripts and relative promoter activity. A** Viral transcripts that encoded the 50 nucleotides upstream of PAS$_2$ were classified as ccc-RNAs. Full-length transcripts lacking this region and terminating downstream of position 2700 were classified as id-RNAs. Shorter Sp1, Sp2 or X regulated transcripts that terminated before position 2700 could not be assigned to either template and are listed as 'unassigned'. Abortive Core associated transcripts (TES at 400 or 2600) were excluded. The stacked bars show the relative abundance of ccc-RNAs, id-RNAs and RNAs of unknown origin in the in vitro (mean values of $n = 3$) and clinical samples with viral read counts shown. **B** Promoter assignments of cccDNA and iDNA transcripts for in vitro (mean values of $n = 3$) and clinical samples with ccc-RNA counts >100. Abortive RNAs (TES: 400 and 2600) and reads of unknown origin were excluded. See Supplementary Fig. 4. Source data are provided as a Source Data file.

transcription can continue into the host genome and generate chimeric RNAs (Fig. 4A). A BLAST search for non-viral sequences (>10 base pairs) within the id-RNAs against the human genome (GRCh38.p13) identified a large number of matches (Fig. 4A). Aligning these chimeric transcripts identified sequences in chromosomes; genes and protein coding loci; intergenic regions, contigs, scaffolds and non-coding DNAs (Supplementary Data 2). The host sequences identified in each of the biopsies were unique and allow us to estimate the number of transcriptionally active integrants (Fig. 4A). Sample 1036 was atypical, with 94% of chimeric transcripts mapping to a single locus (NC_000020.11), suggesting clonal expansion of an integrant bearing hepatocyte (Fig. 4A, Supplementary Data 2). In the remaining samples an average of 390 (219–414) different loci accounted for 95% of the chimeric reads, reflecting the random nature of integration events (Supplementary Data 2). A small population of 'non-chimeric' transcripts were detected, possibly reflecting integration events near host PAS or cryptic/alternative viral PAS usage (Fig. 4A). Very few (<10) 5'host-3'viral chimeric transcripts were found, suggesting that integrant transcription is largely driven by viral promoters, with minimal influence from host regulatory elements. In contrast to the human genome, a BLAST search for chimeric sequences in the cDNA transcriptome (GRCh38.p14) yielded relatively few hits; for example, in biopsy 1030 only 84 chimeric RNAs mapped to the transcriptome, as opposed to 1116 which aligned to the human genome (Fig. 4A). In biopsy 1048, 23% of the chimeric RNAs mapped to the host

transcriptome, the majority being *AC104024.11*, together with 14 further host cDNAs (Fig. 4A). There was limited evidence for conservation of viral integration sites amongst the clinical samples, with each having a unique set of host genes, further supporting a model of random HBV integration.

There was a 5-log range in the length of the chimeric 3' tails (Fig. 4B), with many being thousands of nucleotides long, with implications for RNA half-life, stability and translation[43]. All id-RNAs encoding an S gene ORF included the stop codon at position 2495 and would not generate chimeric fusion proteins. To ascertain whether these id-RNAs are translated, we immunostained an adjacent region of the biopsy for HBs and estimated the frequency of hepatocytes expressing cytoplasmic antigen. Cytoplasmic HBs expression was observed in 10/11 tissue samples, providing compelling evidence that these transcripts are translated (Fig. 4B, Supplementary Fig. 1B)[27]. In contrast, only 4/11 biopsies expressed hepatitis B core antigen (HBcAg), reflecting the low abundance of Core promoter associated transcripts (Supplementary Fig. 1B–C).

Mapping the viral regions of the chimeric RNAs showed a predominance of HBs encoding sequences as detailed in Fig. 2 and highlighted a surprising diversity of 3' termini. Grouping the TES identified unique profiles of id-RNAs in all samples (Fig. 4B–C, Supplementary Fig. 5A). For example, in sample 1036 a single junction accounted for 94% of chimeric RNAs with the majority terminating within a single 25 bp window (Supplementary Fig. 5A). Although DR1 at position 3250 is reported to be the linearisation site during dslDNA genesis, we found many id-RNAs terminated after this motif. We observed many transcripts terminating upstream of this position (Fig. 4C, Supplementary Fig. 5A). In stark contrast, the ccc-RNAs were highly conserved with the majority terminating within a single 25 bp window around PAS$_2$ (Fig. 4C, Supplementary Fig. 5A). There was evidence of a small number of overlength cccDNA derived transcripts that extended beyond PAS$_2$, in agreement with a recent report from Chapus et al.[44]. Despite id-RNAs being relatively rare in vitro these transcripts terminated over a broad range of TES relative to the ccc-RNAs, with minimal evidence for linearisation at DR1 (Fig. 4C). ASOs and siRNA 'translation inhibitor' therapies largely target the DR2-DR1 region at the conserved 3' terminus of id- and ccc- RNAs (Supplementary Fig. 5B). Bepirovirsen is an ASO in clinical trial (B-Well study) that targets the DR2 locus[45] and scanning the HBV transcriptome revealed that up to 40% of id-RNAs lack this motif, providing a natural barrier to functional cure (Supplementary Fig. 5B). This heterogeneity in the 3' termini of id-RNAs gives a new perspective on viral evasion strategies and suggests that transcription from integrated genomes may limit the efficacy of ASO and siRNA therapeutics that target this region.

## Genetic diversity of cccDNA and iDNA derived HBV transcripts

HBV DNA shows up to 5% variation within patients, reflecting the error-prone reverse transcription step of the life cycle that results in genetic polymorphisms with the potential for drug resistance and vaccine escape[46,47]. However, the relative contribution of cccDNA or iDNA to this variability is not known. To address this, we determined the genetic variation of the ccc- and id-RNA populations in the preS2 region (1650–2495) that is common to both templates. To assess intra-biopsy diversity, we generated sample specific consensus sequences as referent genomes for these analyses. Firstly, we assessed polymorphisms in the twenty most abundant host transcripts. These host sequences showed 99.8% similarity, with altered residues randomly distributed and detected only once, reflecting the inherent error rate of RNA Polymerase II[48] and the accuracy of PacBio sequencing (Supplementary Fig. 6A). In contrast, HBV sequences were more polymorphic and annotating the changes showed that ccc-RNA sequences were more variable than id-RNAs, with some individual nucleotides showing as much as 50% variation, as exemplified by samples 1031 and 1035 (Fig. 5A, Supplementary Fig. 6B). HBs mutations previously

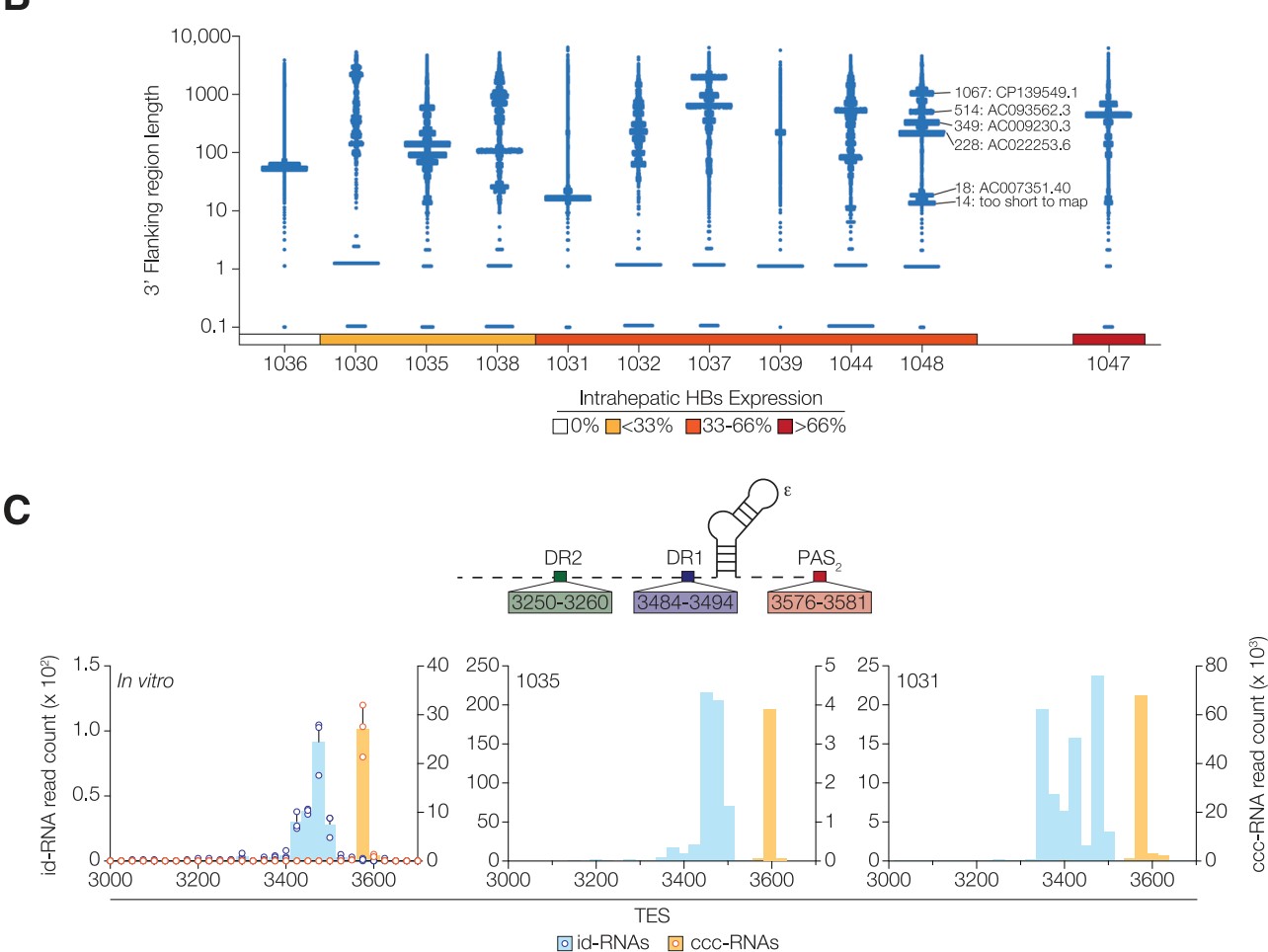

**Fig. 4 | Characterisation of HBV integrant derived transcripts. A** A schematic to show the composition of chimeric transcripts. Reads were aligned to the human genome (GRCh38.p13), and classified as non-chimeric or chimeric, with read counts and most common insertion sites noted. Transcripts were also aligned to the cDNA transcriptome, with the number of chimeric RNAs, as well as the number of individual fusion genes and the most abundant locus tabulated. **B** Scatter plots showing the length of non-viral 3′ flanking regions in chimeric RNAs, with a representative plot from sample 1048 annotated with the most common 3′ flanking regions and their GenBank ID. Clinical samples are presented according to the estimated percentage of HBs expressing hepatocytes in the biopsy. **C** A schematic depicting the 3′ terminal DR1, DR2 and $PAS_2$ motifs and their coordinates. The histograms summarise the 3′ viral junctions of the id-RNAs (blue) and ccc-RNAs (orange) of in vitro ($n = 3$; mean + SD) and representative clinical samples (1035 and 1031). See Supplementary Fig. 5. Source data are provided as a Source Data file.

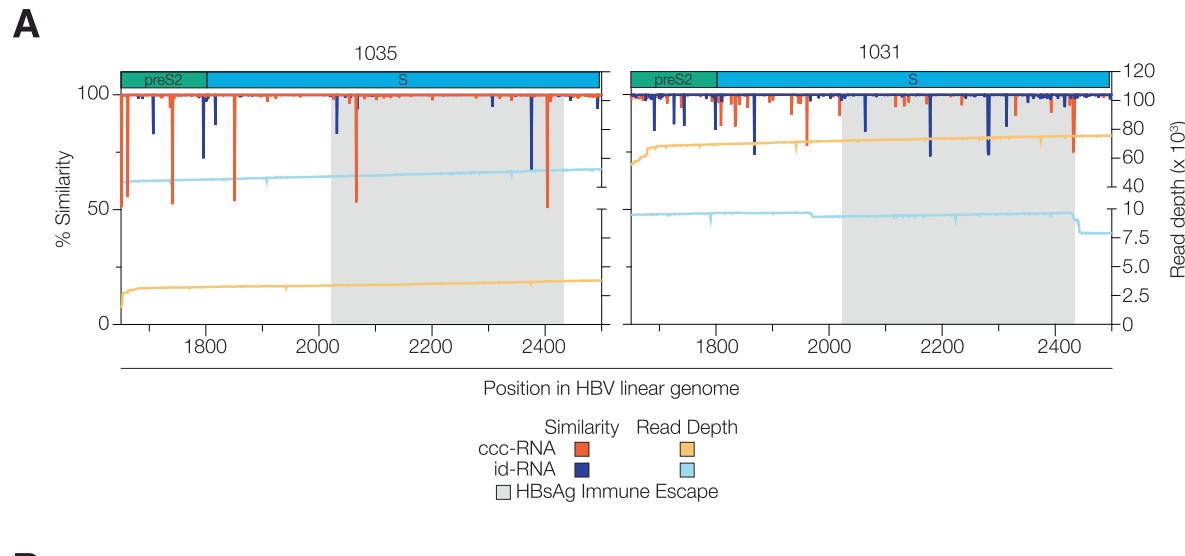

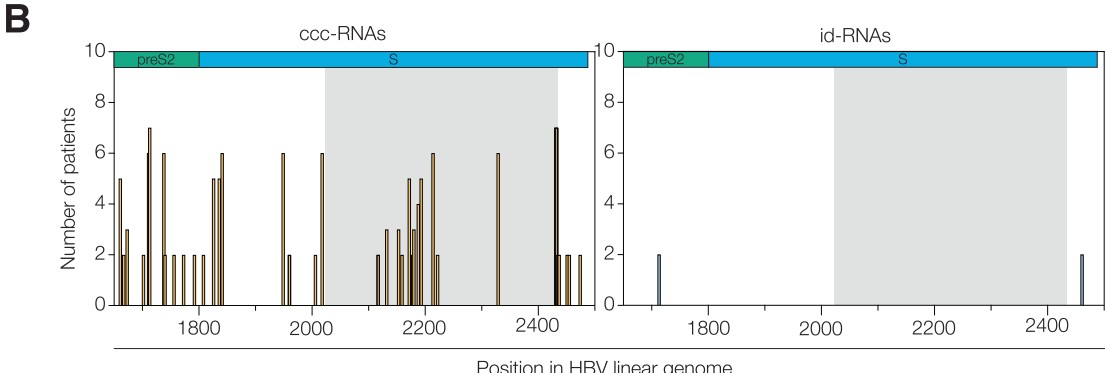

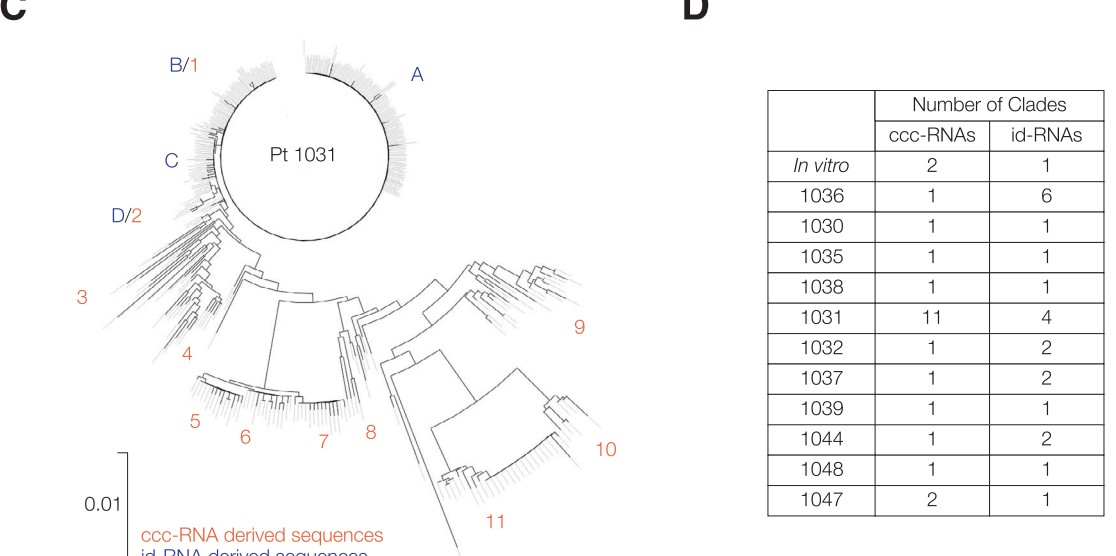

| | Number of Clades | |
|---|---|---|
| | ccc-RNAs | id-RNAs |
| *In vitro* | 2 | 1 |
| 1036 | 1 | 6 |
| 1030 | 1 | 1 |
| 1035 | 1 | 1 |
| 1038 | 1 | 1 |
| 1031 | 11 | 4 |
| 1032 | 1 | 2 |
| 1037 | 1 | 2 |
| 1039 | 1 | 1 |
| 1044 | 1 | 2 |
| 1048 | 1 | 1 |
| 1047 | 2 | 1 |

**Fig. 5 | Genetic diversity of cccDNA and iDNA derived HBV transcripts.**
**A** Sequence similarity of ccc- and id- RNA spanning the 1623 bases of the preS1 ORF are shown as a waterfall plot using independent patient specific consensus sequences (left y-axis). Line traces show the depth of coverage of ccc-RNA (orange) and id-RNA (blue) (right y-axis). Data from clinical samples 1035 and 1031 are shown, with the remaining samples in Supplementary Fig. 6. **B** Frequency distribution of polymorphic residues in ccc- and id- RNA in all clinical samples. Only nucleotides that were polymorphic in at least two samples are shown. Both plots show the preS2 (green) and S (blue) ORFs and the region associated with HBsAg immune escape (grey). **C** A phylogenetic tree to show the relationship between ccc- and id-RNAs in patient sample 1031. Trees were generated to depict the variation in the HBs encoding loci and clades annotated in red and blue, respectively. Trees were generated by cluster analysis (Clusterize package in R) and drawn using the MEGA package. **D** The number of clades identified (IQ-tree) in the ccc- or id- RNA transcriptomes from in vitro and clinical samples. See Supplementary Figs. 6, 7. Source data are provided as a Source Data file.

reported to associate with immune escape[49] occur between positions 2022–2434 and we found this region to be the most variable in the ccc-RNA population (Fig. 5B, Supplementary Fig. 6B), demonstrating that error-prone cccDNA replication constitutes a major source of genetic variation. Analysing an 80 nucleotide region in the HBs-encoding locus allowed us to infer that all patients were infected with the ayw2 strain of HBV, the most prevalent serotype in southern Europe where these samples were collected[50].

To explore the phylogenetic relationship between cccDNA and iDNA we assessed biopsy 1031, the only sample with comparable numbers of ccc-RNA and id-RNAs. Cladistic analysis (IQ-tree) classified these sequences into eleven (1-11) and four clades (A-D), respectively (Fig. 5C). Clade A showed no evidence for linkage with any of the ccc-RNAs, suggesting this iDNA may represent a historical infection. In contrast, ccc-RNA clades 1 and 2 are identical to id-RNA clades B and D, reflecting ongoing active virus replication and recent integration events (Fig. 5C). Although the number of ccc-RNA sequences in the remaining biopsy samples were small, they all showed greater variability than their cognate id-RNAs. Cladistic analysis identified greater numbers of ccc-RNA than id-RNA groups in most samples (Fig. 5D). Consistent with the heterogeneity in chimeric transcripts and inferred integration sites, we observed diverse phylogenies of id-RNAs across our cohort (Supplementary Fig. 7). The in vitro samples had a single clade of id-RNA in all three replicate samples, and 2 ccc-RNA clades that differed by a single nucleotide (Fig. 5D). In summary, we found that id-RNAs dominate the viral transcriptome in chronic disease and are the major source of invariate HBs and HBx antigens that may limit viral escape from therapeutic vaccines or immunoprophylactic approaches.

## Discussion

Targeted long-read sequencing allowed us to profile the hepatic HBV transcriptome in eleven treatment naïve subjects with HBeAg negative chronic disease. Probe enrichment yielded robust sequencing libraries allowing us to resolve canonical and spliced viral RNAs. We found that all viral promoters gave rise to transcripts initiating outside published TSS boundaries, with Sp2 and X promoters yielding the majority of reads that would be under-estimated in short-read or RACE based approaches. Assigning the transcripts to their template of origin allowed us to profile the promoter activities of cccDNA and iDNA. Up to 98% of the viral RNAs were derived from iDNA with Sp2 promoter associated transcripts predominating, consistent with earlier reports[25–27]. iDNA derived transcripts were heterogenous, with both Sp1 and Sp2 promoters driving a repertoire of chimeric virus-host transcripts, truncated transcripts and spliced variants, with many conferring drug resistance to current ASO and siRNA-based therapies. The depth of coverage across the HBs encoding locus allowed us to characterise the phylogenetic relationship between viral episomes and integrants, revealing a link between active viral replication and naturally acquired polymorphism in the absence of therapeutic intervention.

Our study shows the scarcity of full-length genomic RNAs, representing as little as 4% of the viral transcriptome and justifies the probe-enrichment technique[24]. By assigning viral RNAs to both their promoter and template of origin, we defined the promoter activity of episomes and integrants, suggesting different epigenetic regulation of these templates. In these patients we found that pC/pgRNAs were rare, however Core-promoter associated transcripts were detected in all samples providing evidence for an active cccDNA reservoir. Furthermore, abortive and truncated Core-associated transcripts terminating at either PAS$_1$ or a cPAS (positions 400 and 2600 respectively), that may be overlooked by amplicon or short read sequencing methods, were readily detected across the cohort[40]. For example, the cobas HBV RNA assay (Roche Diagnostics) targets the PAS$_2$[51], whilst the dual-target assay from Abbott Laboratories focuses on the 3'-end of core

ORF and 5'-end of X ORF[52]. Of note, these truncated pC/pg RNAs still encode HBeAg and Core ORFs which could explain the reported discordance between HBV RNA serum titres and core related antigen (HBcrAg) expression in some clinical settings[53].

Long-read sequencing allowed us to study the 5' and 3' termini of each HBV transcript, mapping reads to well characterised TSS as well as studying PAS usage, or 3' host flanking regions. HBV TSS were previously identified by precipitating m7G-capped viral transcripts, cloning into sequencing vectors, and amplicon sequencing by Sanger methods[18,19]. As our enrichment strategy is based on sequence homology, there is no bias for capped transcripts, thus degraded or fragmented viral RNAs will be captured and sequenced. When assigning HBV RNAs to previously identified TSS, we found that up to 50% of viral reads were classified as 'unmapped'. These reads were mainly ascribed to the Sp2 and X promoters, with many encoding intact HBs or HBx ORFs and polyA tails, despite their 5' truncation. Long-read sequencing of the viral transcriptomes of human herpesvirus and vaccinia virus in vitro infections also reported large numbers of previously unreported isoforms[54,55]. In contrast, the most abundant host genes showed a high degree of TSS fidelity, suggesting that truncated RNAs were unique to HBV and could be evidence of promiscuous promoter or redirected enhancer activity.

Analysing the 3' flanking regions of the viral reads showed that ~90% of id-RNAs were chimeric, where the 3' termination site of the viral RNA served as the fusion breakpoint between viral and host genomes. The introduction of HBV promoters into the human genome could result in erroneous transcription of genes that would not ordinarily be expressed. A report from Gu et al suggested that EnhII activates the Sp1 and Sp2 promoters in iDNA, consistent with our findings that Sp1 and 2 derived transcripts dominate the viral transcriptome[56]. Our data show that integration events are random, with limited evidence of conserved sites. The rarity of 5'host-3' virus chimeras illustrate that chimeric RNAs derive from active Sp1, Sp2 or X promoters, rather than upstream host promoters and regulatory elements. We should acknowledge that these chimeric transcripts and inferred integration sites are derived from RNA-seq and will not detect transcriptionally silent integrants. Published studies of HBV integration sites have largely used PCR-based platforms and Illumina short read sequencing[57]. One of the most common approaches to map viral integrants is inverse PCR, where the host chromosome is restriction digested and circularised by ligation, opposite-facing HBV primers are then used to prime amplification followed by sequencing[58]. Inverse PCR is not unbiased, as there will be preferential amplification of genomes that integrate near the restriction sites used for library preparation[14]. More recently, oligo-probe enrichment of HBV DNA, in combination with short read sequencing has been employed to identify HBV integrations in Telomerase and MLL4 genes in HBV-associated HCC[59]. Our observations on integration are based on population level, bulk sequencing data. Future studies could employ single cell long-read sequencing methods or spatial transcriptomic approaches[60] to define the HBV infected cell reservoir and to identify the interplay between cccDNA and iDNA bearing hepatocytes in the liver.

Probe enrichment provides the opportunity to detect low copy, novel or rare RNAs from bulk and single cell library preparations. HBx is a regulatory protein that has been ascribed a broad range of functions, including viral replication, acting as a proto-oncogene and potent antagonist of the Smc5/6 restriction factor[61]. However, HBx transcripts are difficult to enumerate using current methodologies, which has limited our understanding of whether HBx can be transcribed from iDNA[14]. HBx RNAs were found in all biopsy samples, including long and short isoforms originating from both cccDNA and iDNA. Further studies are needed to understand the biological role of these HBx isoforms, with potential differences in their ability to trans-complement HBx deficient virus[62].

Enrichment offers the opportunity to study rare or novel viral RNAs, or to map the viral transcriptome in samples with low viral RNA abundance[23]. Several studies have employed long-read sequencing to map Human Papilloma virus breakpoints and integration sites within cervical cancers and in vitro model systems[63,64]; to identify the internal repeats of the latency-associated nuclear antigen of Kaposi's sarcoma-associated herpesvirus[65] and to detect novel spliced RNAs and in-frame truncations in the human herpesvirus and vaccinia virus transcriptomes[54,55,66]. Alternative splicing increases transcript diversity, however the impact on the proteome is not fully understood[67]. The presence of spliced transcripts in HBV infection is a growing area of interest[35] and HBV plasmid transfection studies show the major spliced transcripts are derived from pC or pg RNAs. In contrast, we showed that SP14 and pSP12 spliced isoforms were derived from preS1 RNAs originating from either cccDNA or iDNA. As most studies have identified splice donor:acceptor patterns using short read sequencing[35,68] this may overlook the complexity of HBV splicing and the role of iDNA as a source of spliced RNAs. One limitation of our transcriptome mapping is our focus on HBV genotype D, however, since this represents the second most prevalent genotype worldwide[69] we believe our results are relevant. Our analytical pipeline now provides the opportunity to assess the transcriptome of diverse HBV genotypes in future studies. The depth of coverage our enrichment method provided enabled us to study variation in cccDNA and iDNA derived transcripts. Errors are likely to derive from three sources: (1) host RNA polymerase II mediated transcription; (2) host restriction factor induced errors via RNA editing; (3) erroneous reverse transcription and establishment of error-filled cccDNA. We hypothesised the reverse transcription stages of the HBV life cycle are the most likely source of mutations. Genetic variation in ccc-RNAs may reflect reinfection events with a genetically distinct virus and we cannot formally exclude this possibility. We previously reported that the abundance of id-RNAs was associated with age[27], consistent with integration events increasing over the duration of infection. This leads us to speculate that the genetic distances between ccc- and id- RNAs would increase over time, and this could be reflected in a more complex repertoire of transcripts. Previously reported mutations that associated with HBsAg immune escape[70] were more prevalent in ccc-RNAs. This was reflected in our phylogenetic analysis, where we noted fewer clades in integrant derived transcripts relative to cccDNA. Taken together, our study presents iDNA as a long-term stable reservoir that can express invariant HBs and HBx proteins.

The 3′ TES of id-RNAs were variable; reflecting alternative integration events where some dslDNA terminated upstream of DR1. The archetypal DR1 linearisation site is not strictly observed in vivo, with many integrations resulting from under-length dslDNA fragments. These heterogenous viral 3′ ends pose a hurdle to nucleotide-targeting therapies such as ASOs and siRNAs. Screening the HBV transcriptome showed that >99% of ccc-RNAs encoded the Bepirovirsen motif, whereas 40% of id-RNAs lacked the target motif. Whilst our analysis focussed on Bepirovirsen, several siRNAs and ASOs target this region in the 3′ DR2-DR1 locus, implying a broad mechanism for viral resistance to these new drug candidates.

In summary, targeted long-read sequencing allowed us to profile the activity of the viral promoters with unprecedented detail. Our analytical pipeline assigned reads to a broad range of TSS and identified previously unreported viral RNAs. Our data show ongoing Core promoter activity in the CHB liver, giving rise to canonical cccDNA derived transcripts together with truncated, abortive and overlength RNAs. By interrogating the 3′ termini of all transcripts we were able to determine their template of origin, identifying a common pattern of viral RNAs transcribed from integrants. Assessing the TES identified a new cryptic PAS and the diverse 3′ termini of integrated HBV fragments with alternative breakpoints in all clinical samples. This heterogeneity presents a significant challenge to the identification of conserved targets for novel nucleotide-based therapies. Assessing the phylogeny of integrant and episome derived RNAs showed a relationship between ccc- and iDNA clades, identifying genetically stable integrant lineages, with cccDNA transcription driving natural evolution.

## Methods

All the research presented complies with the ethical regulations as stipulated by King's College Hospital Liver Research Biobank, IRAS project 332608; REC reference 23/LO/0708 and Università del Piemonte Orientale, 28100 Novara REC reference CE90/19. All patients consented for surplus tissue to be used for research purposes.

### Cell culture and de novo infection

HepG2-NTCP cells (gift from Stephan Urban, University of Heidelberg) were maintained on collagen-coated plasticware and cultured in Dulbecco's Modified Eagles Medium (DMEM, Thermofisher) containing Glutamax, 10% foetal bovine serum, 50 U/ml penicillin/streptomycin, and non-essential amino acids. For infection, media was supplemented with 2.5% dimethyl sulphoxide for 72 h. Cells were inoculated with heparin-column purified HBV (GtD, ayw strain)[71], MOI 300 and 4% PEG8000 for 16 h. Viral inoculum was removed, cells washed 3 times in PBS, and infections continued for 6 days, replacing media on day 3. Cells were maintained in 5% $CO_2$, 18% $O_2$.

### Quantitative PCR

Up to 1 μg of RNA was reverse transcribed using a cDNA synthesis kit (PCR Biosystems), following the manufacturer's protocol. Gene expression was quantified using qPCR (SyGreen Blue Mix, PCR Biosystems) with primers specific for the 3′ terminus of all HBV RNAs, referred to as Total HBV RNA, (F: CGGGGCGCACCTCTCTTTA; R:GTGAAGCGAAGTGCACACGG) or specific to the 5′ locus of pC/pgRNA (F:GGGGAACTAATGACTCTAGCTACC; R:TTTAGGCCCCA-TATTAGTGTTGACA), as reported by us and others[17,18,26,72]. Copy numbers were enumerated against an HBV DNA standard curve and adjusted relative to the RPLP0 housekeeper (F:GCAATGTTGC-CAGTGTCTG R:GCCTTGACCTTTTCAGCAA)[17]. RT-qPCR was run on a LightCycler 96 Instrument (Roche), using a program of 95 °C for 2 min followed by 45 cycles at 95 °C for 5 s, 60 °C for 30 s.

### Liver biopsies, RNA extraction and antigen staining

We identified a cohort of 24 patients with CHB who had previously undergone a liver biopsy at King's College Hospital, together with an additional 22 patients from a previous study[27]. Liver biopsy RNAs were extracted from both cohorts; liver tissue was disaggregated in Trizol, and RNAs prepared by phase separation with chloroform and isopropanol as per the manufacturer's instructions[27]. For the in vitro samples, HBV infected HepG2-NTCP cells were lysed in RLT buffer and RNA extracted using the RNeasy kit (Qiagen), following the manufacturer's protocol. To ensure the accurate sequencing of viral RNA all samples were treated with TURBO DNAse I (Thermofisher) to remove DNA as previously reported[24,73]. RNA integrity was assessed using a TapeStation 2100 (Agilent), with a RIN > 7 determining suitability for sequencing. HBV antigens were stained in FFPE liver biopsy sections on a BENCHMARK XT staining system (Ventana Medical Systems) using a commercial streptavidin-biotin technique, with primary antibodies against HBcAg (Leica Biosystems, L1F161) or HBsAg (Santa Cruz Biotechnology, 3E7).

### Targeted long-read sequencing

RNA from 11 CHB biopsy samples from our previous study[27] and 3 in vitro HBV genotype D (ayw strain) infected HepG2-NTCP samples were sequenced. RNA concentration was determined using QuBit, and 150–300 ng of total RNA used for oligo-dT enriched target amplification (Iso-Seq Express Template Preparation system, PacBio), followed by SMART cDNA synthesis (Takara Bio) and HiFi PCR amplification

(Kapa Biosystems). PacBio sample indices (IDT) were added using Iso-Seq Express amplification primers, with unique identifiers for each sample. Quality and quantity of cDNA was assessed using a Bioanalyzer 2100 and a Nanodrop 2000 spectrophotometer (Thermo Scientific). A panel of 74 biotinylated oligonucleotides were used to enrich the HBV RNAs and included 120 bp 'capture' probes that spanned the HBV genome and overlapped with adjacent probes by 60 bp, providing a 2x tiling of all viral RNAs. Probes were designed to bind all HBV genotypes by the inclusion of additional probes to account for genotype specific extensions and deletions, or accounting for >20% divergence in variable regions[23,24]. Oligos were incubated with cDNA at 65 °C for 4 h, captured with streptavidin Dynabeads at 65 °C for 45 minutes followed by washing (xGen Hybridisation kit, IDT). Capture libraries were pooled and samples ligated with SMRTbell adaptors, according to the manufacturer's instructions. Samples were sequenced on an 8 M SMRT Cell using a Sequel II system (CCS protocol)[24,74].

## Mapping HBV transcripts

HiFi libraries were demultiplexed using the PacBio SMRTLink software to assign the sequences obtained according to their sample index. Polyadenylated sequences were aligned to a HBV ayw genotype D3 genome using minimap2[75] (GenBank ID NC_003977.2) permuted to begin at the pC TSS[19] through to the PAS$_2$, corresponding to the longest HBV transcript and referred to as D3L[19,35]. For reference to previous reports, the EcoRI restriction site (GAATTC) would be at nucleotide position 1658, with the A at 1659 representing the first base of NC_003977.2[19]. This viral referent was appended to the human cDNA transcriptome (Homo_sapiens.GRCh38.cdna.all.fa.gz, ensembl.org) and sequences filtered to remove reads with MAPQ scores of less than 30. CIGAR strings were extracted from the BAM file and parsed using the GenomicAlignments package in R[76]. The TSS for all reads aligning to the HBV genome were classified by assigning the 5' terminus to one of the following co-ordinates (bp): pC-L: 0–13; C-AT1: 12–211; pC-S: 212–279; pgRNA: 284–308; C-AT2: 280–1196; preS1: 1278–1302; Sp1-AT: 1197–1572*; preS2: 1572–1640; S: 1641–1697; Sp2-AT: 1697–2405; X: 2406–2430 and X-AT: 2431–3581. The AT designations define alternative TSS outside the boundaries of previously reported TSS, with * denoting the Sp1-AT-region (1,97–1277 + 1303–1572) that flanks the preS1 TSS. Reads were assigned to their nearest upstream promoter with: Core: 3388–3410 (EcoRI: 1730–1752); Sp1: 1196–1276 (EcoRI: 2720–2800); Sp2: 1436–1536 (EcoRI: 2960–3260) and X: 2888–2928 (EcoRI: 1230–1270), as previously defined (Supplementary Fig. 2)[19]. Spliced transcripts were identified and mapped where intron junctions were within 12 bases of previously reported donor:acceptor sites[35]. To ensure inferences on viral transcripts were based on authentic transcripts and not DNA contaminants, only polyadenylated reads were analysed. Splice patterns were enumerated using the Needleman Wunsch and Smith Waterman algorithms with either the R Biostrings package or the EMBL Emboss server and visualised using ggSashimi[77].

## 3′ Termini mapping

To discern between integrant and cccDNA derived transcripts, the 3' terminus of each HBV read was assessed for the presence of the canonical viral PAS. Reads were interrogated for the 50 bp region that ends in the viral PAS sequence (TATAAA), with a 90% match required. If present the read was labelled as ccc-RNA, or as id-RNA if not. Reads terminating at the cPAS at ~2600 could originate from either cccDNA or iDNA. To account for minor sample-sample variation, we allowed for a 100 bp tolerance, and identified any read terminating after position 2700 and lacking the 3' PAS motif as id-RNA. HBV transcripts were assessed for the presence of motifs of interest, for example Bepirovirsen target-sequence (5′-TGCAGAGGTGAAGCGAAGTGC-3′ at 3261–3241)[45]. The TSS and HBV read length were extracted from the CIGAR strings, with their sum defining the transcription end site (TES). Where HBV transcripts had a non-viral flanking sequence, soft clipped regions ( >10 bp in length) were extracted, and mapped to the human transcriptome. The Rsubread package identified chimeric transcripts, with soft clipped regions aligned the GRCh38.p14 cDNA transcriptome using minimap2. The length, locus, and frequency of flanking regions are annotated (Supplementary Data 2).

## Assessing polymorphism and phylogeny

Genetic polymorphism was assessed from an alignment of the S2-S region of HBV (1685–2590) that is present in both ccc- and id- RNAs, using patient specific consensus sequences. For each nucleotide position we calculated similarity scores based on the consensus sequence. Trees were built and clades identified with IQ-tree[78], using the SYM + R3 model, selected by the modeltest algorithm in Model Finder[79].

## Statistics

Statistical tests were carried out in Prism 10 with all tests listed in their respective figure legends.

## Reporting summary

Further information on research design is available in the Nature Portfolio Reporting Summary linked to this article.

## Data availability

All data presented in the main and Supplementary Figs. are provided as a Source Data file. The sequencing data generated in this study have been deposited in the NCBI database under accession codes PRJNA1225648 and PRJNA1227884. Source data are provided with this paper.

## Code availability

All original code is deposited on Zenodo (https://doi.org/10.5281/zenodo.16539674).

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

## Acknowledgements

We would like to thank Geoff Dusheiko, Ivana Carey and Kosh Agarwal (Kings College Hospital) for provision of samples and their invaluable guidance on this study; Stephan Urban (University of Heidelberg) for supplying HepG2-NTCP cells, Jochen Wettengel and Ulrike Protzer (Technical University of Munich) for purified HBV stocks; Azim Ansari (University of Oxford) for biotinylated probes. Research in the McKeating laboratory is funded by the Wellcome Trust (Investigator Award 200838/Z/16/Z and Discovery Award 225198/Z/22/Z) and the Chinese Academy of Medical Sciences (CAMS) Innovation Fund for Medical Science (CIFMS), China 2024-I2M-2-001-1. James Lok is funded by an MRC clinical research training fellowship MR/Y001168/1. Esther Ng is supported by the Centre for Human Genetics, University of Oxford.

## Author contributions

All authors have approved the submitted version of the manuscript and agree to be accountable for the integrity and accuracy of the study. They have contributed as follows: Investigation: J.M.H., J.L., A.M. and S.T.; Data Curation: J.M.H. and P.B.; Resources: J.L., N.W., A.M., S.T., Y.W., D.J.; Software: E.N., P.B., B.E.; Analysis: J.M.H., P.B.; Conceptualisation: J.A.M.; Supervision: J.A.M.; Writing M.S.: J.M.H., J.A.M., P.B.; Review & editing M.S.: All authors.

## Competing interests

The authors declare no competing interests.
