## [Peer Review file · Nature Communications]

Mapping episomal and integrated hepatitis B transcriptomes in chronically infected liver biopsies uncovers heterogeneity with the potential for drug-resistance.

Corresponding Author: Professor Jane McKeating

Version 0:

Reviewer comments:

Reviewer #1

(Remarks to the Author)

(1) General comments

HBV infection poses a significant global public health challenge, with a notable absence of curative treatments. The complexity of the HBV transcriptome presents a primary obstacle to developing such treatments. However, our understanding for the HBV transcriptome remains limited. To address this issue, the authors utilized a targeted long-read sequencing method to analyze the hepatic HBV transcriptome in eleven HBeAg-negative liver samples and three in vitro samples derived from HBV-infected hepatoma cells. Through sequencing libraries enriched using probes, they discovered numerous genomic and sub-genomic transcripts originating from cccDNA and iDNA, encompassing previously unidentified spliced, truncated, and chimeric viral RNAs. By assigning these viral transcripts to their respective DNA templates, the authors characterized the differential promoter activities of cccDNA and HBV integrants. Additionally, they demonstrated the genetic diversity of cccDNA- and iDNA-derived HBV transcripts. In summary, the authors have dissected the patterns of the HBV transcriptome, with their findings shedding light on the potential implications for the efficacy of several nucleotide-targeting drug candidates. Nonetheless, the findings and conclusions of this study may lack universality and representativeness due to the exclusive use of genotype D samples and the omission of infection duration as a consideration. It is plausible that different HBV genotypes and prolonged infection could yield distinct HBV transcriptome patterns. Consequently, the technique validation and proof of concept presented in this study are deemed insufficient. These flaws in this study must be rectified, otherwise the study's conclusions and the importance of its findings would suffer severely.

(2) Specific comments:

a)Major :

1.Regarding RNA extraction, despite the authors' utilization of DNase to eliminate any contaminating DNA, there remains a possibility that DNA residues, encompassing both host DNA and HBV DNA, could persist and potentially serve as templates during subsequent amplification process. Therefore, it is imperative for the authors to furnish supplementary and credible experimental evidence to unequivocally demonstrate the complete removal of all DNA isoforms.

2.The authors stated that all subjects included in the study were treatment-naïve individuals with HBeAg-negative chronic disease. However, Supplementary Figure 1 presents contrasting information, revealing that at the time of biopsy, there were 2 HBeAg+ and 2 HBeAg- were undergoing antiviral treatment. This raises questions regarding the authors' sample selection process and the specific inclusion and exclusion criteria employed. Could the authors please clarify how they chose the samples and provide a detailed explanation of the criteria used for both inclusion and exclusion?

3.Why did the authors limit their study exclusively to samples of genotype D? Do you consider the transcriptome patterns observed in genotype D to be representative of those in other HBV genotypes?

4.Although the Supplementary information provided details on the disease stages of liver fibrosis and necro-inflammatory activity, careful considerations were not given to disease stage and infection duration in the study design. Furthermore, the in vitro samples were infected with HBV for a mere 6 days. Have you given thought to the possibility that different disease stages and longer durations of infection might result in distinct transcriptome patterns of HBV?

b)Minor :

1.Page 15 line 518-519: The forward primer designed specifically for Total HBV RNA (F: ACGGGGCGCACCTCTTTA) was

incorrect; it should actually be ACGGGGCGCACCTCTCTTTA. How did this error occur, and what are the implications for the quantitation of Total HBV RNA?

2. Page 4 line 107-108: The authors should provide clarification on the methodological mechanism they employed to distinguish between total HBV transcripts and pgRNA. Similarly, they need to elucidate how they differentiated among various isoforms of HBV transcripts, namely pC-L, C-AT1, pC-S, pgRNA, and C-AT2.

3. Page 5 line 163-169: It would be beneficial to provide a table that summarizes and compares previously reported TSS with newly discovered TSS.

4. Page 6 line 191-192: The utilization of the PAS2 motif to distinguish RNAs derived from cccDNA (ccc-RNA) and integrant-derived RNAs (id-RNA) is questionable. To ensure accuracy, this dataset should be cross-verified with datasets of both non-chimeric and chimeric transcripts, with the latter being considered more reliable.

5. Page 8 line 246-249: The original results mentioned here ought to be included in the supplementary information. Additionally, conducting an association analysis between id-RNA and peripheral HBsAg would be beneficial. Furthermore, have you examined the expression of HBeAg in these biopsy samples?

6. Page 11 line 368: "bearing cells," should be "bearing cells."

7. Figure 2B: If this schematic were presented in a circular format, with the addition of amplification direction and genome location, it would enhance its readability.

8. Figure 4D: Please clarify the meaning of the asterisk (*) indicated in the table on the right?

(Remarks on code availability)

Reviewer #2

(Remarks to the Author)

I read the manuscript submitted to Nature Communications with great interest. In this study, the authors focus on an interesting topic in the HBV field: how cccDNA and integrated (i)DNA contribute to the genesis of HBV transcripts. Using a targeted long-read PacBio sequencing, they identified the viral transcriptome in liver biopsies from treatment-naïve CHB patients, instead of cell or primate models. This approach provides an overview of HBV transcriptional complexity and new scientific concepts, including that iDNA is the major resource of subgenomic id-RNAs, whereas cccDNA generates, in general, 3'-terminal longer transcripts. Additionally, the authors identified new transcripts with non-canonical 3' ends. Taken together, this data partially explains the non-responsiveness of current RNA-targeting therapies and helps further drug discovery. Here are my arguments:

Major comments:

(1) Lines 117-120, the authors aligned all sequences to a single HBV reference, NC_003977.2. However, those patients were originally infected by different HBV strains, I assume. According to the PacBio sequencing raw data, it is more reasonable to assemble the 3.5kb overlength genome of each sample and ascertain the maximal likelihood HBV reference individually (likely choose the one sequence with the highest number of reads). Some information gets inaccurate if all sequences are aligned to the NC_003977.2.

(2) The cryptic PAS (cPAS) at c.a 2600 bp is an interesting finding. Is it conserved in all HBV genotypes? Has it been mutated in some HBV strains? What about the function?

(3) Lines 246-249, visualising HBsAg by immunostaining is an important piece of this story. It is worth presenting this data from all eleven tissue samples, regardless of negative or positive staining, in a supplementary figure to fully support Fig.4B.

(4) Lines 546-547, it was not clear to understand the 74 biotinylated oligonucleotides. Given c.a. 30 probes with 120 bp can cover the 3.5kb RNA genome, does the excessive number of oligonucleotides equally contribute to the whole genome, or do they all target the least conserved regions among genotypes? Provide more information.

(5) The direct and exact evidence of cccDNA and iDNA existence was not shown in the manuscript. It was difficult to judge whether a DNA fragment in the sequencing data comes from an episome or an integrant. Also, both transcriptional activity and the number of cccDNA and iDNA can determine the amount of transcripts. How about the number of cccDNA per cell and iDNA per cell in all eleven liver biopsies, determined by specific PCR methods?

Minor comments:

(1) Lines 155-158, if a 5' primer binds the epsilon region and a 3' primer does not bind the shorter transcripts, they could not be amplified at all when the PCR product is designed to be longer and they cannot pose a problem in pgRNA quantification using PCR approaches.

(2) Lines 212-213, how to explain the difference of RNA splicing in vitro and in vivo (Supplementary Fig.3B), in the latter of which one TPM was more often and the pattern was different from in vitro.

(3) Lines 556-557, except genotypes, it would be better to list serotypes adw, adr, ayw, and ayr for all patients used in this study.

(4) Any evidence supporting or denying a second HBV reinfection that leads to cccDNA diversity (Fig.5C)?

(5) Showing the cPAS region in the scheme would help reading (Fig.1B)

(6) Some typos in e.g. line 66 and line 408.

(Remarks on code availability)

Reviewer #3

(Remarks to the Author)

This manuscript describes the mapping of the Hepatitis B virus transcriptome using probe enrichment and long read sequencing. The experimental approach used has been previously published using in vitro data only. This study leads on from the previous study by applying this method to patient samples. Samples were from liver biopsies from HBeAg negative untreated patients. The entire transcriptome from all viral promoters was compared across patients and to in vitro cell culture samples. The viral transcripts were identified as originating from either the cccDNA or the integrated DNA, and the transcription start, and end sites were mapped. The integrated chimeric transcripts were identified to characterise HBV insertion into host sequences.

The methods resulted in identification of novel transcription start and end sites for the viral transcripts, with noted differences between the in vitro samples and the patient samples. The authors noted a cryptic polyadenylation site (PAS) that has not been previously described. Confirming previous reports, most of the viral transcripts from the patient biopsies were from integrated HBV forms. Analysis of the integrations revealed random integration events. Several spliced viral transcripts were also detected. Analysis of genetic diversity showed most diversity comes from the cccDNA form, not from integrated forms. The novel methods utilised have allowed a comprehensive in-depth analysis of the viral transcriptome from patient liver samples. The results are very interesting and important to the field, the analysis and identification of these novel viral transcripts will have implications for the efficacy of new HBV therapies currently under investigation.

The manuscript is very well written, and in general the data has been well presented and the conclusions reached are valid. We have some minor comments, suggestions and clarifications outlined below.

Specific comments:

The title is about "drug-resistant lineages" detected in the viral transcripts. However, the results only focus on a single drug (Bepirovirsen) and these results are relegated to the supplementary figures. If the focus of the paper includes drug-resistance, could the authors include data for additional therapies (eg. siRNA targeting locations etc.)?

The HBV numbering used needs to be better explained. It would be helpful if the position of the first nucleotide of the preCore transcript which is designated as the first base was better defined compared to standard HBV numbering.

Figure 1B. The numbering in Figure 1B is not accurate. The distance between DR1 to PA1 is indicated as 368 nucleotides, whereas the distance indicated between the DR1 and PAS2 is 316 nucleotides. This distance should be identical as it covers exactly the same sequence? Also, the HBx open reading frame should encompass both DR2 and DR1. The discrepancy in the numbering in this figure makes it difficult to interpret the transcription start and end sites identified in this study.

Line 148. The authors indicate a cryptic PAS (cPAS) at ~2600bp. Can the authors identify any potential sequence motif for a PAS at this position?

In figure 1E, for the patient samples, the majority of the transcripts appear to terminate at PAS2 (except for the pC-S transcripts). In the text (Lines 152-154) the authors state that many core-associated transcripts including pgRNA terminated at PAS1? This statement needs clarifying.

Figure 1E, could the diagram include each of the isoforms – pC-L, pC-S etc. for clarity?

Figure 2B. Could the figure legend provide more detail on how to interpret these types of graphs? Please indicate what the blips represent.

Lines 166-167. "Elucidating the TSS of the viral RNAs allowed us to identify their cognate promoters (Supplementary Fig. 2A)." Could the authors please specify how they have identified the promoters?

Line 211. Supplementary Table 1 should be referenced with Supplementary Fig. 3A when mentioning unique patterns of spliced viral RNAs.

Line 278. The text states the host sequences show 99.8% similarity, however in the Supplementary Figure 5A legend this value is 99.5%?

Line 284. The reference for potential immune escape HBs mutations is not appropriate (Ref 50, Olinger et al – only refers to genotype E variants). Possibly a review such as Echevarria and Avellon, J Med Virol DOI 0.102/jmv.20605?

Lines 295-296. The authors comment that Supplementary Fig 6 shows the numbers of cccRNA and integrated RNA, however only integrated RNA forms are shown in the figure.

Figure 3 legend. "Full-length transcripts lacking this region and terminating downstream of position 2,700 were classified as id-RNAs." Why is the position 2,700 important here? This is not defined in the methods to differentiate between cccDNA and integrant transcripts?

Supplementary Figure 3A. The 1031 graph should be 2000 on the Y axis, not 20,000.

Line 212-213. The authors state "Higher levels of the SP1 spliced RNA were found in samples with detectable pgRNA (Supplementary Fig. 3B), consistent with previous studies." (Reference 42- Sozzi et al). However, reference 42 examined in vitro cell culture samples, not patient samples. This needs correction.

(Remarks on code availability)

Reviewer #4

(Remarks to the Author)

(Remarks on code availability)

Version 1:

Reviewer comments:

Reviewer #1

(Remarks to the Author)

The authors have adequately addressed all my questions and appropriately acknowledged the study's limitations in the revised manuscript. However, the text still contains numerous language and writing issues that require careful correction prior to publication.

1. Page 4 line 125: sequences were aligned, using
2. Page 4 line 146: quantified pC Long and short isoforms
3. Page 4 line 148: (C-AT1: TSS range 13-211), in Fig. 1C TSS range 12-211
4. Format of units and numbers: 3.5 kb vs. 3.5kb

(Remarks on code availability)

Reviewer #3

(Remarks to the Author)

Thank you for the response to our comments. We are happy that most of these have been appropriately corrected. However, we still have some queries to clarify.

Specific comments 1. Thank you for including the references to additional drugs in clinical trials. We understand the confidentiality issues that preclude examination of the exact sequences. We apologise if our comment was unclear, but we do not think the title of the paper should include "drug-resistant lineages" when the only data referring to drug resistance is in the supplementary data. Perhaps the title should be re-worded to point out that drug resistant lineages were not uncovered, just the potential for this was identified.

Specific comments 2. Thank you for including the annotation for the EcoR1 site on Figure 1B. However, there is still some clarification required regarding the numbering. The distance from DR1 to the polyA site should be around 92 nucleotides (nt) in most genotypes, including the reference NC_003977.2. If the EcoR1 site in this genome is at nt 1524, then the DR1 site should be nt 167-177 and the poly A is at nt 259-264. This numbering is different to that noted in Figure 1B? In addition, in Supp Figure 1E, the pgRNA TSS is denoted at 296 +/- 12 for both this paper and Reference 19. However, in Reference 19 the pgRNA TSS is approx. 10nt upstream of DR1, but DR1 is indicated in Figure 1B as nt 68-78? Could the authors please further clarify the position and numbering for these elements in Figure 1B and Supp Figure 1E?

Specific comments 3. Regarding point 9 from our previous comments, Supplementary Table 1 should be referenced with Supplementary Fig. 4A when mentioning unique patterns of spliced viral RNAs (line 235).

(Remarks on code availability)

Reviewer #5

(Remarks to the Author)

I was asked by the editorial team of Nature Communications to fill in as mediating referee for reviewer #2, who was not available to review the revised manuscript. As mediator, I was asked to evaluate the responses to the previous comments from reviewer #2.

The authors have addressed most of the concerns raised in the first round. The responses to the major comments 1-4 and to the minor comments were satisfactory. The extended texts improve the manuscript.

Regarding comment 2 on the cryptic PAS (cPAS), it would be interesting to investigate other genotypes in follow-up studies, from patient samples or in cell culture, to see if the cPAS is conserved in different genotypes.

Regarding comment 5, the authors have provided information on cccDNA positivity of the liver biopsies. Determination of cccDNA numbers and numbers of HBV DNA integrations is still missing.

Additional comments:

1. I concur with reviewer #3's comment 1. The title of the manuscript should be reviewed and potentially revised. The manuscript does not focus on drug-resistant lineages.
2. Citation 5 should be updated (e.g. with DOI: 10.1016/j.jhep.2025.03.018).

(Remarks on code availability)

Reviewer #1**(1) General comments**

HBV infection poses a significant global public health challenge, with a notable absence of curative treatments. The complexity of the HBV transcriptome presents a primary obstacle to developing such treatments. However, our understanding for the HBV transcriptome remains limited. To address this issue, the authors utilized a targeted long-read sequencing method to analyze the hepatic HBV transcriptome in eleven HBeAg-negative liver samples and three in vitro samples derived from HBV-infected hepatoma cells. Through sequencing libraries enriched using probes, they discovered numerous genomic and sub-genomic transcripts originating from cccDNA and iDNA, encompassing previously unidentified spliced, truncated, and chimeric viral RNAs. By assigning these viral transcripts to their respective DNA templates, the authors characterized the differential promoter activities of cccDNA and HBV integrants. Additionally, they demonstrated the genetic diversity of cccDNA- and iDNA-derived HBV transcripts. In summary, the authors have dissected the patterns of the HBV transcriptome, with their findings shedding light on the potential implications for the efficacy of several nucleotide-targeting drug candidates. Nonetheless, the findings and conclusions of this study may lack universality and representativeness due to the exclusive use of genotype D samples and the omission of infection duration as a consideration. It is plausible that different HBV genotypes and prolonged infection could yield distinct HBV transcriptome patterns. Consequently, the technique validation and proof of concept presented in this study are deemed insufficient. These flaws in this study must be rectified, otherwise the study's conclusions and the importance of its findings would suffer severely.

Response: We thank the reviewer for asking questions about the role of infecting viral genotype and duration of infection on viral transcriptome mapping. We have provided some text along with citations on HBV genotypes and the limitations of epidemiological studies linking genotype to disease phenotype (Lines 42-45). Chronic hepatitis B is commonly first diagnosed when patients present with late stage liver disease and hence the duration of infection is frequently unknown. We justify our selection of clinical samples from HBV genotype D infections, the second most common genotype worldwide, that allows comparison and validation with the frequently used infectious molecular clone used for our in vitro infection. Finally, we state in the discussion the potential limitations of studying genotype D virus and the importance of using our analytical pipeline to study diverse HBV strains in the future (Lines 417-421).

Specific comments:

1. Regarding RNA extraction, despite the authors' utilization of DNase to eliminate any contaminating DNA, there remains a possibility that DNA residues, encompassing both host DNA and HBV DNA, could persist and potentially serve as templates during subsequent amplification process. Therefore, it is imperative for the authors to furnish supplementary and credible experimental evidence to unequivocally demonstrate the complete removal of all DNA isoforms.

Results: We isolated and generated our sequencing libraries using oligo-dT purification to isolate polyadenylated RNAs and include new data (Supplementary Fig.3C) along with associated text in the Results (Lines 194-198) and Methods (Lines 613-614) to explain this. In addition, all samples were subjected to DNase digestion before library preparation (Lines 568-570).

2. The authors stated that all subjects included in the study were treatment-naïve individuals with HBeAg-negative chronic disease. However, Supplementary Figure 1 presents contrasting information, revealing that at the time of biopsy, there were 2 HBeAg+ and 2 HBeAg- were undergoing antiviral treatment. This raises questions regarding the authors' sample selection process and the specific inclusion and exclusion criteria employed. Could the authors please clarify how they chose the samples and provide a detailed explanation of the criteria used for both inclusion and exclusion?

Response: We apologise for this omission and have provided new text to justify our sample selection (Lines 118-120) along with a new Supplementary Fig.1B that provides metadata for each of the cases.

3. Why did the authors limit their study exclusively to samples of genotype D? Do you consider the transcriptome patterns observed in genotype D to be representative of those in other HBV genotypes?

Response: As described above we sought to explore the transcript patterns in a subgroup of patients who were treatment naïve and disease stage matched and selected a group of 11 HBeAg negative patients and have listed their clinical characteristics in Supplementary Fig.1B. We have provided new text to discuss HBV genotypes (Lines 42-45) and the potential limitations of our study that focuses on HBV genotype D (Lines 417-421).

4. Although the Supplementary information provided details on the disease stages of liver fibrosis and necro-inflammatory activity, careful considerations were not given to disease stage and infection duration in the study design. Furthermore, the in vitro samples were infected with HBV for a mere 6 days. Have you given thought to the possibility that different disease stages and longer durations of infection might result in distinct transcriptome patterns of HBV?

Response: To address the first point we selected patients in the same disease classification according to the EASL disease staging who were treatment naïve at time of biopsy (Lines 118-120, Supplementary Figure 1B). We list necro-inflammation and liver fibrosis scores for the cases studied and are actively investigating the link between viral infection and disease pathology using single cell tissue transcriptomic approaches. To address part two of this point, we provide our rationale along with citations for sampling the in vitro infections after 6 days that reflects ongoing cccDNA transcription with limited viral integration events (Lines 130-132). As most chronic hepatitis B cases are diagnosed at a late stage of infection it is not possible to accurately infer duration of infection.

Minor :

1. Page 15 line 518-519: The forward primer designed specifically for Total HBV RNA (F: ACGGGGCGCACCTCTTTA) was incorrect; it should actually be CGGGGCGCACCTCTCTTTA. How did this error occur, and what are the implications for the quantitation of Total HBV RNA?

Response: we have corrected the primer sequences that was a transposition error whilst compiling the methods (Line 551). We can assure the reviewer we used the forward primer as listed in the revised MS, as per our earlier publication (D'Arienzo et al. 2019).

2. Page 4 line 107-108: The authors should provide clarification on the methodological mechanism they employed to distinguish between total HBV transcripts and pgRNA. Similarly, they need to elucidate how they differentiated among various isoforms of HBV transcripts, namely pC-L, C-AT1, pC-S, pgRNA, and C-AT2.

Response: We have provided further details and citations on the PCR methodology for enumerating HBV pgRNA and total viral transcripts in the Methods (Lines 551-554). In brief, pgRNA can be amplified by primers targeting a unique region before the TSS of preS1. This loci is targeted for pgRNA detection in many published studies and we provide relevant citations. Total RNAs are quantified by primers that target the conserved 3' terminal that is shared between all viral transcripts. We provide new text in the methods (Line 604) to explain our designation of the viral transcripts that is based on 5' TSS and 3' TES mapping.

3. Page 5 line 163-169: It would be beneficial to provide a table that summarizes and compares previously reported TSS with newly discovered TSS.

Response: We thank the reviewer for this suggestion and provide a new Supplementary Fig.2E to show how our nomenclature simplifies the classification HBV transcripts and provide a comparison with previously published TSS.

4. Page 6 line 191-192: The utilization of the PAS2 motif to distinguish RNAs derived from cccDNA (ccc-RNA) and integrant-derived RNAs (id-RNA) is questionable. To ensure accuracy, this dataset

should be cross-verified with datasets of both non-chimeric and chimeric transcripts, with the latter being considered more reliable.

Response: We have provided further citations to justify the use of DR1 as the linearisation site for dsDNA genesis and subsequent integration (Lines 208-209). We cite studies from our lab and others that have used this region for classifying ccc- and id- RNAs

5. Page 8 line 246-249: The original results mentioned here ought to be included in the supplementary information. Additionally, conducting an association analysis between id-RNA and peripheral HBsAg would be beneficial. Furthermore, have you examined the expression of HBeAg in these biopsy samples?

Response: as per the reviewer's suggestion, we discuss and present HBs and HBc scoring information (Lines 264-268, Supplementary Fig. 1B-C). The cases selected for sequencing are a subset of a larger published cohort where the relationship between peripheral and hepatic viral markers was studied (Magri et al. 2022). For diagnostic purposes HBeAg is routinely measured in serum as a peripheral biomarker and liver biopsies are not stained for this antigen. To the best of our knowledge we are not aware of antibodies specific for HBeAg that are validated to stain human FFPE tissue sections.

6. Page 11 line 368: "bearing cells," should be "bearing cells."

Response: Corrected (Line 393).

7. Figure 2B: If this schematic were presented in a circular format, with the addition of amplification direction and genome location, it would enhance its readability.

Response: We thank the reviewer for this suggestion and have provided a cartoon depicting a circular genome in Supplementary Fig. 2A. This has been annotated to show both HBV transcripts and translated products, as well as the position of DR1, DR2 and the PAS.

8. Figure 4D: Please clarify the meaning of the asterisk (*) indicated in the table on the right?

Response: We believe this in reference to Fig. 1D and Fig. 2A. As such, we have clarified the text in the Methods section (Line 608) along with a new Supplementary Fig. 2 and associated legend, where TSS classifications are described. In short, this reflects where the canonical TSS boundary for preS1 is flanked by the Sp1-associated transcript region.

Reviewer #2

I read the manuscript submitted to Nature Communications with great interest. In this study, the authors focus on an interesting topic in the HBV field: how cccDNA and integrated (i)DNA contribute to the genesis of HBV transcripts. Using a targeted long-read PacBio sequencing, they identified the viral transcriptome in liver biopsies from treatment-naïve CHB patients, instead of cell or primate models. This approach provides an overview of HBV transcriptional complexity and new scientific concepts, including that iDNA is the major resource of subgenomic id-RNAs, whereas cccDNA generates, in general, 3'-terminal longer transcripts. Additionally, the authors identified new transcripts with non-canonical 3' ends. Taken together, this data partially explains the non-responsiveness of current RNA-targeting therapies and helps further drug discovery. Here are my arguments:

Major comments:

1. Lines 117-120, the authors aligned all sequences to a single HBV reference, NC_003977.2. However, those patients were originally infected by different HBV strains, I assume. According to the PacBio sequencing raw data, it is more reasonable to assemble the 3.5kb overlength genome of each sample and ascertain the maximal likelihood HBV reference individually (likely choose the one sequence with the highest number of reads). Some information gets inaccurate if all sequences are aligned to the NC_003977.2.

Response: We have expanded the text in both the Results and Methods sections to clarify whether the D3L reference genome or patient specific consensus sequences were used (Lines 122-124 and 296-298, respectively). The D3L referent simplifies the mapping of key genetic features and transcripts, whereas the patient specific consensus sequences were required to identify polymorphisms.

2. The cryptic PAS (cPAS) at c.a 2600 bp is an interesting finding. Is it conserved in all HBV genotypes? Has it been mutated in some HBV strains? What about the function?

Response: We have amended the Results section (Lines 158-160) to describe our scanning of the 2,600 region for previously reported PAS hexamers, with a relevant citation. Given the absence of these hexamer motifs we were unable to search published HBV sequences for homologous signatures. The conserved nature of Core promoter associated transcripts terminating in this region in both clinical and experimental samples suggests a cccDNA dependency that we are currently investigating.

3. Lines 246-249, visualising HBsAg by immunostaining is an important piece of this story. It is worth presenting this data from all eleven tissue samples, regardless of negative or positive staining, in a supplementary figure to fully support Fig.4B.

Response: We provide new text and data showing patient by patient scoring of HBsAg biopsy stains (Lines 264-268, Supplementary Fig.1B). These patients represent a subgroup of cases from a larger previously published cohort (Magri et al. 2022) and since these HBsAg stained tissue images are published we are unable to duplicate in the current paper. However, we include new images showing HBcAg staining to illustrate the scoring criteria (Supplementary Fig.1C).

4. Lines 546-547, it was not clear to understand the 74 biotinylated oligonucleotides. Given c.a. 30 probes with 120 bp can cover the 3.5kb RNA genome, does the excessive number of oligonucleotides equally contribute to the whole genome, or do they all target the least conserved regions among genotypes? Provide more information.

Response: We provide additional text in the methods section on the probe design that aligns with the manufacturers recommended 2x tiling method (Lines 584-588). In addition the probes were designed to be HBV genotype agnostic, with potential for diagnostic application.

5. The direct and exact evidence of cccDNA and iDNA existence was not shown in the manuscript. It was difficult to judge whether a DNA fragment in the sequencing data comes from an episome or an integrant. Also, both transcriptional activity and the number of cccDNA and iDNA can determine the amount of transcripts. How about the number of cccDNA per cell and iDNA per cell in all eleven liver biopsies, determined by specific PCR methods?

Response: Please see earlier response to reviewer 1 point 1 where we describe our rationale and method for sequencing polyadenylated RNAs, that will preclude the sequencing of viral or host DNA. As part of our earlier study we published hepatic cccDNA estimates for the full cohort and have included these data for the cases sequenced in new Supplementary Figure 1B.

Minor comments:

1. Lines 155-158, if a 5' primer binds the epsilon region and a 3' primer does not bind the shorter transcripts, they could not be amplified at all when the PCR product is designed to be longer and they cannot pose a problem in pgRNA quantification using PCR approaches.

Response. We thank the reviewer for this comment and have revised the text to address this point (Lines 167-168)

2. Lines 212-213, how to explain the difference of RNA splicing in vitro and in vivo (Supplementary Fig.3B), in the latter of which one TPM was more often and the pattern was different from in vitro.

Response: we have included new text to address this point along with a citation that addresses the importance of splicing as a means of increasing transcript diversity with applications beyond virology (Lines 411-412).

3. Lines 556-557, except genotypes, it would be better to list serotypes adw, adr, ayw, and ayr for all patients used in this study.

Response: Thanks for this excellent suggestion and by analysing the S gene for the serotype defining residues identified by Norder et al (1992), we established that all patients are infected with the ayw2 serotype (Lines 308-310).

4. Any evidence supporting or denying a second HBV reinfection that leads to cccDNA diversity (Fig.5C)?

Response: This is an excellent point that we have discussed in the revised MS (Lines 426-427). In brief, we cannot exclude superinfection with a similar HBV serotype and have listed this as a potential limitation in the discussion.

5. Showing the cPAS region in the scheme would help reading (Fig.1B)

Response: We have annotated the cPAS region as suggested in Fig. 1B.

6. Some typos in e.g. line 66 and line 408.

Response: We apologise for any spelling mistakes and have carefully proof-read the revised MS.

Reviewer #3

This manuscript describes the mapping of the Hepatitis B virus transcriptome using probe enrichment and long read sequencing. The experimental approach used has been previously published using in vitro data only. This study leads on from the previous study by applying this method to patient samples. Samples were from liver biopsies from HBeAg negative untreated patients. The entire transcriptome from all viral promoters was compared across patients and to in vitro cell culture samples. The viral transcripts were identified as originating from either the cccDNA or the integrated DNA, and the transcription start, and end sites were mapped. The integrated chimeric transcripts were identified to characterise HBV insertion into host sequences. The methods resulted in identification of novel transcription start and end sites for the viral transcripts, with noted differences between the in vitro samples and the patient samples. The authors noted a cryptic polyadenylation site (PAS) that has not been previously described. Confirming previous reports, most of the viral transcripts from the patient biopsies were from integrated HBV forms. Analysis of the integrations revealed random integration events. Several spliced viral transcripts were also detected. Analysis of genetic diversity showed most diversity comes from the cccDNA form, not from integrated forms. The novel methods utilised have allowed a comprehensive in-depth analysis of the viral transcriptome from patient liver samples. The results are very interesting and important to the field, the analysis and identification of these novel viral transcripts will have implications for the efficacy of new HBV therapies currently under investigation. The manuscript is very well written, and in general the data has been well presented and the conclusions reached are valid. We have some minor comments, suggestions and clarifications outlined below.

Specific comments:

1. The title is about “drug-resistant lineages” detected in the viral transcripts. However, the results only focus on a single drug (Bepirovirsen) and these results are relegated to the supplementary figures. If the focus of the paper includes drug-resistance, could the authors include data for additional therapies (eg. siRNA targeting locations etc.)?

Response: Given the commercial nature of ASO and siRNA drugs in clinical trials we have been unable to identify their exact sequences, however, they are known to target the DR1 region and we

have revised Supplementary Fig.5B to list six additional candidates (AHB-137; AB-729; VIR-2218; ARC-520 si74, ARC-520 si77 and JNJ-3989). Since the Bepirovirsen target is publically available we were able to assess motif conservation in all patients.

2. The HBV numbering used needs to be better explained. It would be helpful if the position of the first nucleotide of the preCore transcript which is designated as the first base was better defined compared to standard HBV numbering.

Response: We permuted the numbering scheme to provide a more meaningful alignment, such that position 1 represents the first base of the longest HBV transcript (pC-Long) as published by ourselves and others (Ng et al 2023, Lim et al 2021). We have annotated the EcoRI site on the linear genome/transcript schematic in Fig.1B (position 1,524) and provide a circular genome map in Supplementary Fig. 2A. In addition, we have extended the Methods section to help clarify this point and provided an 'EcoRI decode' for the viral promoters (Lines 609-611).

3. Figure 1B. The numbering in Figure 1B is not accurate. The distance between DR1 to PA1 is indicated as 368 nucleotides, whereas the distance indicated between the DR1 and PAS2 is 316 nucleotides. This distance should be identical as it covers exactly the same sequence? Also, the HBx open reading frame should encompass both DR2 and DR1. The discrepancy in the numbering in this figure makes it difficult to interpret the transcription start and end sites identified in this study.

Response: We thank the reviewer for spotting this error and have corrected Fig. 1B in the revised MS.

4. Line 148. The authors indicate a cryptic PAS (cPAS) at ~2600bp. Can the authors identify any potential sequence motif for a PAS at this position?

Response: A similar comment was made by reviewer 2 (point 2) – please see our response above.

5. In figure 1E, for the patient samples, the majority of the transcripts appear to terminate at PAS2 (except for the pC-S transcripts). In the text (Lines 152-154) the authors state that many core-associated transcripts including pgRNA terminated at PAS1? This statement needs clarifying.

Response: We have revised the text and clarified our interpretation of these data (Lines 163-166).

6. Figure 1E, could the diagram include each of the isoforms – pC-L, pC-S etc. for clarity?

Response: We have revised the cartoon in Fig. 1E to identify both the 5' and 3' ends of each transcript, and how this was used for transcript assignment.

7. Figure 2B. Could the figure legend provide more detail on how to interpret these types of graphs? Please indicate what the blips represent.

Response: We have added some further details to the legend of this figure to help the reader interpret these data.

8. Lines 166-167. "Elucidating the TSS of the viral RNAs allowed us to identify their cognate promoters (Supplementary Fig. 2A)." Could the authors please specify how they have identified the promoters?

Response: We have extended the text (Lines 609-611), and provide a new Supplementary Fig 2E to clarify this point. Briefly, we used the coordinates of the promoters as previously described to assign each TSS to their nearest upstream promoter.

9. Line 211. Supplementary Table 1 should be referenced with Supplementary Fig. 3A when mentioning unique patterns of spliced viral RNAs.

Response: We have referenced as suggested.

10. Line 278. The text states the host sequences show 99.8% similarity, however in the Supplementary Figure 5A legend this value is 99.5%?

Response: We have checked our data and the correct figure is 99.8% and have revised the legend.

11. Line 284. The reference for potential immune escape HBs mutations is not appropriate (Ref 50, Olinger et al – only refers to genotype E variants). Possibly a review such as Echevarria and Avellon, J Med Virol DOI 0.102/jmv.20605?

Response: BIG thanks for this insightful comment, we have revised the citation and checked that our immune escape region depicted in the figure corresponds to the new reference.

12. Lines 295-296. The authors comment that Supplementary Fig 6 shows the numbers of cccRNA and integrated RNA, however only integrated RNA forms are shown in the figure.

Response: We have corrected the text as appropriate (Lines 320-322).

13. Figure 3 legend. “Full-length transcripts lacking this region and terminating downstream of position 2,700 were classified as id-RNAs.” Why is the position 2,700 important here? This is not defined in the methods to differentiate between cccDNA and integrant transcripts?

Response: We have expanded the description in the Methods section (Lines 622-625). We note that cPAS using transcripts could originate from either ccc- or i- DNA. As such, we allowed a 100bp window and designated any read terminating after 2,700 that lacked a 3’PAS motif as id-RNA.

14. Supplementary Figure 3A. The 1031 graph should be 2000 on the Y axis, not 20,000. Line 212-213. The authors state “Higher levels of the SP1 spliced RNA were found in samples with detectable pgRNA (Supplementary Fig. 3B), consistent with previous studies.” (Reference 42- Sozzi et al). However, reference 42 examined in vitro cell culture samples, not patient samples. This needs correction.

Response: We have corrected the figure as suggested, and replaced the Sozzi reference with one from Van Buuren, where samples were derived from patient samples.

Reviewer #4

Response: We’d like to take the opportunity to thank Nature Communications for running this initiative. Training the next round of expert peer reviewers is a valuable endeavour that should be applauded.

We look forward to a favourable decision on our revised manuscript and to seeing our paper *in press* in Nature Communications.

Reviewer #1: The authors have adequately addressed all my questions and appropriately acknowledged the study's limitations in the revised manuscript. However, the text still contains numerous language and writing issues that require careful correction prior to publication.

Page 4 line 125: sequences were aligned ,using;

Page 4 line 146: quantified pC Long and short isoform;

Page 4 line 148: (C-AT1: TSS range 13-211), in Fig.1C TSS range 12-211

Format of units and numbers: 3.5 kb vs. 3.5kb

Response: We've corrected and tracked the editorial items listed in the revised MS

Reviewer #3: Thank you for the response to our comments. We are happy that most of these have been appropriately corrected. However, we still have some queries to clarify.

Specific comments 1. Thank you for including the references to additional drugs in clinical trials. We understand the confidentiality issues that preclude examination of the exact sequences. We apologise if our comment was unclear, but we do not think the title of the paper should include "drug-resistant lineages" when the only data referring to drug resistance is in the supplementary data. Perhaps the title should be re-worded to point out that drug resistant lineages were not uncovered, just the potential for this was identified. **Response:** We've revised the title of our study to 'Mapping episomal and integrated hepatitis B transcriptomes in chronically infected liver biopsies uncovers heterogeneity with the potential for drug-resistance'.

Specific comments 2. Thank you for including the annotation for the EcoR1 site on Figure 1B. However, there is still some clarification required regarding the numbering. The distance from DR1 to the polyA site should be around 92 nucleotides (nt) in most genotypes, including the reference NC_003977.2. If the EcoR1 site in this genome is at nt 1524, then the DR1 site should be nt 167-177 and the poly A is at nt 259-264. This numbering is different to that noted in Figure 1B? In addition, in Supp Figure 1E, the pgRNA TSS is denoted at 296 +/- 12 for both this paper and Reference 19. However, in Reference 19 the pgRNA TSS is approx. 10nt upstream of DR1, but DR1 is indicated in Figure 1B as nt 68-78? Could the authors please further clarify the position and numbering for these elements in Figure 1B and Supp Figure 1E? **Response:** We would like to thank the reviewer for their expert advice on the numbering of these key motifs within the HBV genome. We identified a mislabelling of the DR1 and DR2 motifs in Fig.1B and have corrected the schematic in the revised manuscript. As requested, we have revised the boundaries of pC-L and C-AT1 so they do not overlap and have reannotated the EcoRI position in Supplementary Figure 1E and Figure 1B, respectively.

Specific comments 3. Regarding point 9 from our previous comments, Supplementary Table 1 should be referenced with Supplementary Fig. 4A when mentioning unique patterns of spliced viral RNAs (line 235). **Response:** We have cited Supplementary Table 1 in the tracked revised MS.

Reviewer #5: I was asked by the editorial team of Nature Communications to fill in as mediating referee for reviewer #2, who was not available to review the revised manuscript. As mediator, I was asked to evaluate the responses to the previous comments from reviewer #2. The authors have addressed most of the concerns raised in the first round. The responses to the major comments 1-4 and to the minor comments were satisfactory. The extended texts improve the manuscript.

Regarding comment 2 on the cryptic PAS (cPAS), it would be interesting to investigate other genotypes in follow-up studies, from patient samples or in cell culture, to see if the cPAS is conserved in different genotypes. **Response:** We agree and are currently identifying patients with diverse viral infections for long-read DNA and RNA-sequencing.

Regarding comment 5, the authors have provided information on cccDNA positivity of the liver biopsies. Determination of cccDNA numbers and numbers of HBV DNA integrations is still missing. **Response:** As our study sequenced RNA we cannot enumerate cccDNA copies or determine the frequency of integration events. In the discussion we discuss the limitations of our study and list future plans to source liver DNA and RNA for sequencing both templates.

Additional comments:

I concur with reviewer #3's comment 1. The title of the manuscript should be reviewed and potentially revised. The manuscript does not focus on drug-resistant lineages. **Response:** We agree and have revised our MS title.

Citation 5 should be updated (e.g. with DOI: 10.1016/j.jhep.2025.03.018). **Response:** We have updated citation 5 and this is tracked in the revised MS.

We thank the reviewers for their time in providing constructive comments on our study that have helped to strengthen our manuscript and we look forward to seeing our work in press.